# Integrated molecular and functional characterization of the intrinsic apoptotic machinery identifies therapeutic vulnerabilities in glioma

Elizabeth G. Fernandez[1], Wilson X. Mai[1], Kai Song[2], Nicholas A. Bayley[1], Jiyoon Kim[3], Henan Zhu[1], Marissa Pioso[1], Pauline Young[1], Cassidy L. Andrasz[1,9], Dimitri Cadet[1], Linda M. Liau[4,5], Gang Li[3], William H. Yong[6], Fausto J. Rodriguez[6], Scott J. Dixon[7], Andrew J. Souers[8], Jingyi Jessica Li[3,4,9,10,11], Thomas G. Graeber[1,4,12,13], Timothy F. Cloughesy[1,4,14] & David A. Nathanson[1,4] ✉

Genomic profiling often fails to predict therapeutic outcomes in cancer. This failure is, in part, due to a myriad of genetic alterations and the plasticity of cancer signaling networks. Functional profiling, which ascertains signaling dynamics, is an alternative method to anticipate drug responses. It is unclear whether integrating genomic and functional features of solid tumours can provide unique insight into therapeutic vulnerabilities. We perform combined molecular and functional characterization, via BH3 profiling of the intrinsic apoptotic machinery, in glioma patient samples and derivative models. We identify that standard-of-care therapy rapidly rewires apoptotic signaling in a genotype-specific manner, revealing targetable apoptotic vulnerabilities in gliomas containing specific molecular features (e.g., *TP53* WT). However, integration of BH3 profiling reveals high mitochondrial priming is also required to induce glioma apoptosis. Accordingly, a machine-learning approach identifies a composite molecular and functional signature that best predicts responses of diverse intracranial glioma models to standard-of-care therapies combined with ABBV-155, a clinical drug targeting intrinsic apoptosis. This work demonstrates how complementary functional and molecular data can robustly predict therapy-induced cell death.

Next-generation sequencing has revealed the altered genetic and transcriptional landscape of human cancers[1,2]. These data have led to the molecular classification of tumours, and to subsequent tailored treatment regimens that have successfully linked genetic profiles to therapeutic vulnerabilities[3,4]. However, even in the presence of a validated oncogenic driver, molecularly guided therapy often fails to elicit clinical responses in cancer patients[5,6]. The array of genetic alterations within a given tumour, coupled with the high plasticity of signaling networks, can provide tumours with unpredictable avenues to circumvent therapeutic interventions[7,8].

Functional precision medicine (FPM) attempts to address these challenges by assessing the response of live cells to specific perturbations[9]. When tumourigenic pathways have defined measurable outputs, perturbations (e.g., drugs, metabolites, peptides) that

alter the functional circuitry are added to live cells to assess the state of that specific pathway[9]. This approach can identify tumour vulnerabilities as well as predict drug responses, without pre-existing knowledge of the molecular alterations present in the cell[10–12]. Notably, the application of FPM to match cancer patients with specific therapies has shown significant clinical promise[13–16]. However, these approaches have typically required deriving ex vivo cell cultures of patient cells for drug screening where, in particular for solid tumours, the non-native environment can preclude the establishment of a viable or faithful cell culture[17–19].

BCL-2 homology domain (BH3) profiling is a functional assay that interrogates tumour intrinsic apoptotic pathway function without the need to establish an ex vivo cell culture[20–22]. Briefly, by exposing a single-cell suspension of tumour cells to pro-apoptotic BH3-only peptides that have specific affinities for various anti-apoptotic BCL-2 family proteins (e.g., MCL-1, BCL-$X_L$ and BCL-2), both a tumour's dependency on an apoptotic block(s) and the overall apoptotic potential for cell death (i.e., mitochondrial priming) can be assessed[23]. As a result, BH3 profiling can predict in vivo sensitivity to conventional anti-cancer agents (DNA-damaging and targeted therapies)[20,24], and to characterise and guide the translation of small molecule inhibitors that directly target the apoptotic blocks (e.g., BH3 mimetics), some of which have been clinically evaluated (e.g., Venetoclax, Navitoclax)[25]. Thus, unlike other FPM methods that necessitate the establishment of ex vivo cell cultures, BH3 profiling offers a robust and direct method to functionally assess a critical cell fate pathway in cancer cells freshly isolated from tumour samples[26].

Here, we hypothesise that integrating genomic and BH3 functional profiling may reveal specific genetic events that associate with particular apoptotic dependencies. In this investigation, we focus on malignant glioma, a cancer characterised by both genetic diversity and profound resistance to cell death[26,27]. Using patient tumour specimens, together with a library of patient-derived gliomaspheres and intracranial mouse xenografts to confirm and expand on our clinical findings, our integrated molecular and BH3 functional analysis reveals previously unrecognized insight into apoptotic resistance in glioma leading to a tailored therapeutic approach for gliomas with specific genetic and functional features.

## Results
### BCL-$X_L$ and MCL-1 create a dual apoptotic barrier in glioma
Prior work in a small number of cell culture models suggests glioblastoma (GBM) cells may have survival dependencies on the apoptotic blocks BCL-$X_L$, BCL-2 and/or MCL-1[28–30]. However, the composition of apoptotic block dependencies among glioma patient tumours − in which the genomic landscape is diverse − has yet to be functionally defined. To search for relationships between recurring genetic alterations and specific apoptotic block dependencies in glioma patients, we molecularly and functionally characterised 30 unique patient glioma tumours by performing whole exome (WES) and RNA sequencing (RNA-seq) coupled with BH3 profiling (Fig. 1A, Supplementary Data Table 1). Purified tumour cell specimens, which included freshly isolated patient tumour cells from both newly diagnosed and recurrent *IDH* WT ($n = 20$) GBM as well as *IDH* mutant gliomas ($n = 10$), underwent BH3 profiling and molecular analysis following tissue dissociation and removal of red blood cells (RBCs), myelin and CD45+ cells (Fig. 1A). Molecular analyses revealed our patient tumour samples contained the commonly recurring genetic events in glioma as described by The Cancer Genome Atlas (TCGA)[31] (Fig. 1B).

To functionally define the apoptotic block(s) dependencies among our patient sample cohort, ~$1.5 × 10^6$ purified patient cells were permeabilized and the mitochondria were exposed to either individual or combinations of peptides or small molecule inhibitors that have specific affinities for the anti-apoptotic blocks: BCL-$X_L$ (peptide: HRK),

MCL-1 (peptide: MS1), BCL-2 (BH3 mimetic: ABT-199), and dual BCL-2/BCL-$X_L$ (peptide: BAD) (Fig. 1A). Consistent with previous work and as demonstrated in Ext Fig. 1A using established cell line controls, the induction of cytochrome *c* release by specific peptide(s) − as measured by flow cytometry − indicates a reliance on a particular apoptotic block to inhibit mitochondria outer membrane permeabilization (MOMP) and, thereby, prevent intrinsic apoptosis[20]. Interestingly, across our patient samples, BH3 profiling revealed that the most significant increase in cytochrome *c* release, irrespective of genetic alterations, was observed only when peptides specific for BCL-$X_L$ and MCL-1 were combined (Fig. 1B, C). Thus, BCL-$X_L$ and MCL-1 act as cross-compensatory blocks to prevent glioma tumour MOMP and potentially apoptosis.

To test the prediction that inhibition of both MCL-1 and BCL-$X_L$ is required to promote glioma apoptosis, we utilized a library of molecularly diverse patient-derived gliomaspheres ($n = 26$), which retain the genetic features and tumour initiating potential of GBM patient tumours (Supplementary Data Fig. 1B)[18]. Like the patient tumour cohort, BH3 profiling of gliomaspheres revealed dual BCL-$X_L$ and MCL-1 blocks to prevent MOMP (Fig. 1C and Supplementary Data Fig. 1B). In agreement with this finding, only the combination of small molecule drugs targeting BCL-$X_L$ (A-1155463) and MCL-1 (S63541) were consistently capable of inducing apoptosis in the panel of gliomaspheres (Fig. 1D−F). Therefore, BH3 profiling of primary GBM cells can strongly predict apoptotic block dependencies (Fig. 1G).

Intriguingly, while this functionally defined dependency on BCL-$X_L$ and MCL-1 is coupled to their high RNA expression in patients and gliomaspheres[28] (Supplementary Fig. 1C), this relationship did not extend to protein levels of BCL-$X_L$ and MCL-1, whose expression, along with other BCL-2 family members, was variable among patient-derived gliomaspheres and glioma patient samples (Supplementary Fig. S1D, E). These findings confirm previous studies showing that protein levels of anti-apoptotic BCL-2 family members are often incapable of predicting apoptotic block dependencies[32]. Taken together, these data indicate BCL-$X_L$ and MCL-1 comprise a dual apoptotic barrier to block intrinsic apoptosis across molecularly heterogeneous patient glioma tumours.

### Standard-of-care therapy results in an exclusive survival dependency on BCL-$X_L$ in *TP53* wild-type GBM
Ionizing radiation (IR) is standard treatment for GBM patients; however, tumour responses are typically short-lived[33]. Resistance to cell death via apoptosis may play a causal role in the lack of treatment durability[34]. Indeed, while IR administration at a range of doses strongly reduced gliomasphere proliferation, it weakly induced cell death across the entire panel of cells tested ($n = 17$) (Fig. 2A). To examine the role of BCL-$X_L$ or MCL-1 (or both) in impeding treatment-mediated apoptosis, we exposed a small cohort of gliomaspheres ($n = 8$) to IR and subsequently quantified therapy induced changes in the functional dependencies on BCL-$X_L$ and MCL-1 using the treatment-incorporating approach of Dynamic BH3 Profiling (DBP)[35]. DBP revealed acute IR treatment (48 hours) caused a subset of gliomasphere lines to shift to a single BCL-$X_L$ dependency to prevent MOMP while the remaining gliomaspheres maintained their dual BCL-$X_L$ and MCL-1 blocks (Fig. 2B).

To detect potential associations between glioma genotypes and changes in apoptotic dependencies with radiotherapy, we queried whole-exome data from our gliomaspheres. All gliomaspheres in which IR induced a sole dependency on BCL-$X_L$ for survival were characterised as wild-type *TP53* (*TP53* WT) (Fig. 2C). This singular BCL-$X_L$ dependency was demonstrated by caspase-dependent intrinsic apoptosis when BCL-$X_L$ was targeted genetically or pharmacologically (with A-1155463) in combination with IR (Fig. 2C, D, Supplementary Fig. S2A, B). By contrast, the gliomaspheres maintaining their dual dependency on BCL-$X_L$ and MCL-1 under IR treatment had inactivating mutations in

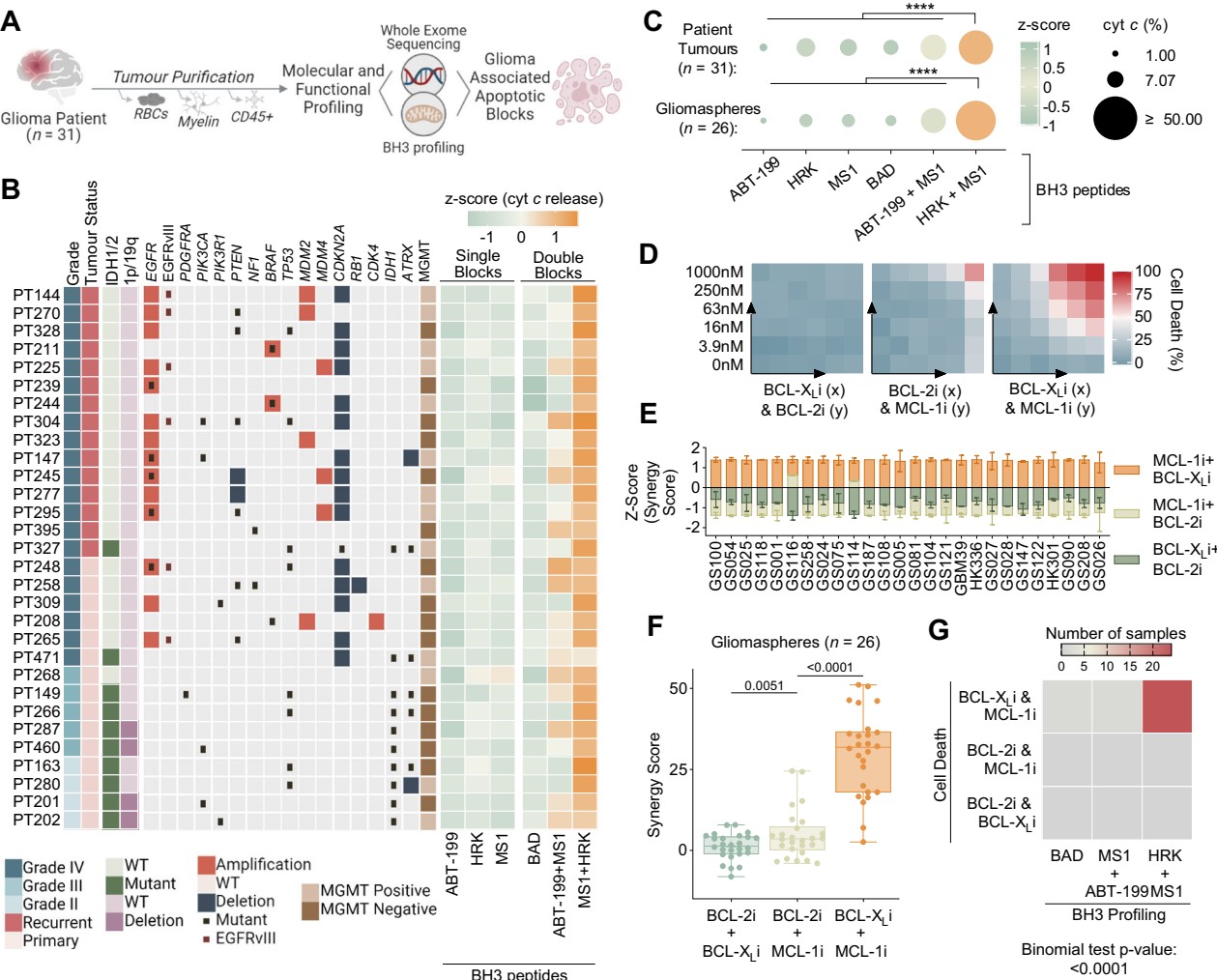

**Fig. 1 | Multiomic characterisation of the intrinsic apoptotic machinery in glioma patient and patient-derived samples.** **A** Workflow describing tumour purification and subsequent molecular (whole-exome sequencing) and functional (BH3 profiling) characterisation of glioma patient samples ($n = 31$). Also see Supplementary Table 1. **B** Heat maps describing patient sample clinical characteristics, copy number alterations and mutations of gene frequently alerted in GBM and BH3 profiling of the apoptotic blocks. **C** BH3 profiling is plotted as a z-score across the sample. Peptide concentrations are as follows: ABT-199: 1 μM (anti BCL-2), MS1: 10 μM (anti MCL-1), HRK: 100 μM (anti BCL-$X_L$), BAD: 10 μM (anti BCL-2 and BCL-$X_L$). BH3 profiling of anti-apoptotic blocks in patient tumours and gliomaspheres. Dot plot displays % cytochrome c release as dot size and z-score as dot color ($n = 31$ and $n = 26$, two-tailed, paired t test compares HRK + MS1 to all conditions). Also see Supplementary Fig. 1B. **D** Example heatmaps of cell viability (Cell Titer Glo) after

48 hours. of treatment with combinations of ABT-199 (BCL-2i), A-1155463 (BCL-$X_L$i), and S63856 (MCL-1i) in gliomaspheres GS028. BH3 mimetics concentrations are as follows: 0 nM, 3.9 nM, 15.6 nM, 62.5 nM, 250 nM, 1000 nM. **E** Bar graphs (mean ± s.d.) of zip synergy scores plotted as a z-scores for gliomaspheres ($n = 26$). Synergy scores calculated from cell viability (Cell Titer Glo) experiments with combinations of BH3 mimetics: ABT-199 (BCL-2i), A1155463 (BCL-$X_L$i), and S63856 (MCL-1i) (mean ± s.d., $n = 2$ experimental replicates). **F** Box and whiskers plot (mean, hinges at 25th and 75th percentiles, ± min to max) of synergy scores for each gliomasphere ($n = 26$), grouped by combinations of BH3 mimetics (two-tailed, paired t test). **G** Heat map displays summarizes highest scoring combination of cell death and BH3 profiling data for the 26 gliomaspheres. Statistics calculated using the binomial test. $p > 0.05$ = ns; $p < 0.05$ = *$p < 0.01$ = **$p < 0.001$ = ***$p < 0.0001$ = ****. See also Table S1 and Figure S1.

*TP53* (i.e., hereafter referred to as mut-p53). These gliomaspheres were insensitive to IR combined with BCL-$X_L$ inhibition (Fig. 2C, D, Supplementary Figure S2B). In an orthogonal approach, CRISPR-Cas9 mediated *TP53* knock-out (KO) prevented two p53 WT lines (HK301 and GS025) from converting to a single dependency on BCL-$X_L$ with IR (Fig. 2E) and subsequent cell death (Fig. 2F). Likewise, pharmacological p53 transcription inhibition with pifitrin-α (PFTα)[36] mitigated cell death with combined IR and BCL-$X_L$i in p53 WT gliomaspheres (Supplementary Fig. 2C). Finally, the IR-induced switch to BCL-$X_L$ dependence preceded the induction of GBM cell senescence (Supplementary Fig. 2D)[37], indicating that IR treatment can create a BCL-$X_L$ dependency independent of GBM cell senescence. Collectively, these results demonstrate that all GBM tested have dual BCL-$X_L$ and MCL-1 anti-apoptotic blocks; however, in the face of acute IR treatment, p53 promotes an exclusive dependency on BCL-$X_L$ for tumour cell survival.

To investigate whether IR promotes a BCL-$X_L$ dependency in a p53-dependent manner in vivo, we established orthotopic brain xenografts from p53 WT (GS025) and mut-p53 (GS005) gliomasphere lines. Once exponential tumour growth was confirmed, mice were treated with 10 Gy IR cranially and 48 h later tumour cells were extracted, purified (blood, myelin, and mouse cells were removed), and immediately used for DBP. Consistent with our in vitro results, IR created a single BCL-$X_L$ dependency only in p53 WT GBM tumour xenografts (Fig. 2G). Moreover, only in the p53 WT xenograft could BCL-$X_L$ knockdown augment IR tumour growth inhibition (Fig. 2H).

Our findings show IR treatment induces a single dependency on BCL-$X_L$ through a p53-dependent mechanism. We were curious whether this therapy-induced effect on the GBM intrinsic apoptotic machinery was limited to IR. The alkylating agent temozolomide (TMZ)

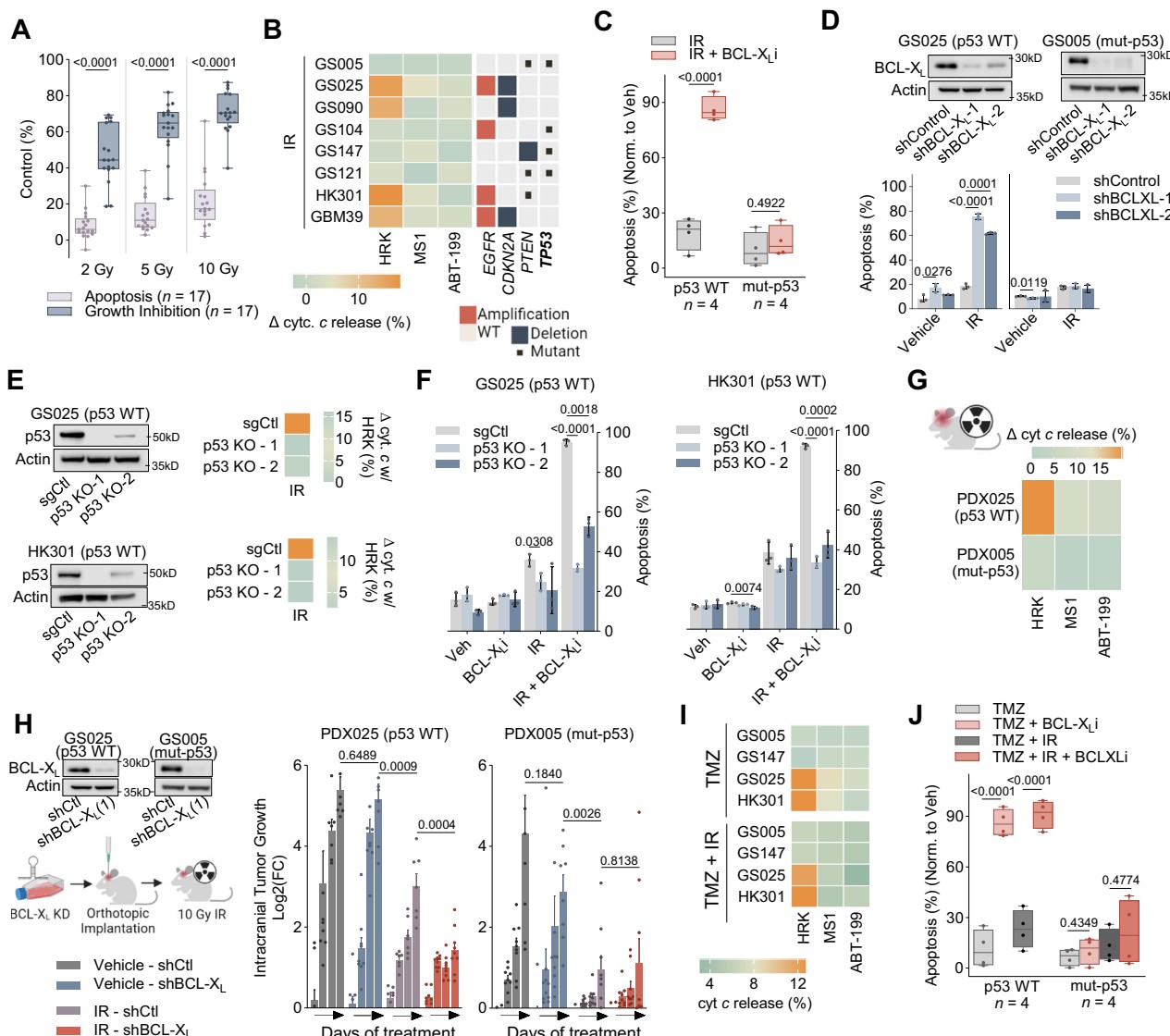

**Fig. 2 | IR creates an exclusive survival dependency on BCL-X$_L$ in p53 wild-type GBM. A** Box plots of radiation induced growth inhibition and apoptosis, each dot represents an individual gliomasphere and the mean of three biological replicates (mean ± s.d., two tailed, paired t test). **B** DBP post IR treatment of gliomaspheres. Heatmap displays copy number and mutations for *EGFR*, *CDKN2A*, *PTEN* and *TP53*. **C** Box plots show apoptosis post IR and/or BCL-X$_L$i. Each dot represents a gliomasphere. Data were normalized to vehicle (median ± quartiles, two-tailed, unpaired t test). **D** shRNA used to reduce expression of BCL-X$_L$. Apoptosis assessed after treatment with IR (mean ± s.d., unpaired t test with Welch correction, *n* = 3 biological replicates). **E** CRISPR guides were used to KO p53. DBP of KOs after IR treatment, shows change in precent cytochrome *c* release with HRK (mean ± s.d., unpaired t test with Welch correction, *n* = 2 biological replicates). **F** Apoptosis evaluated in p53 KOs, 5 days after treatment with BCL-X$_L$i and/or IR (mean ± s.d.,

unpaired t test with Welch correction, *n* = 3 biological replicates). **G** Heat map shows change in precent cytochrome *c* release between mice irradiated with 10 Gy and untreated mice with HRK, MS1 and ABT-199 (*n* = 2 independent replicates). **H** shRNA KD of BCL-X$_L$ before cells were orthotopically implanted and irradiated with 10 Gy IR 3 days post injection. Bar plots display log2(fold-change) of tumour burden over time (mean ± SEM, *n* = 9 mice). Grouped comparisons made were made with two-tailed, unpaired t tests. **I** DBP post TMZ or TMZ + IR treatment of gliomaspheres. **J** Box plots show apoptosis with TMZ or TMZ + IR in combination with BCL-X$_L$i (mean ± s.d., two tailed, paired t test). IR (5gy), TMZ (50 μM), BCL-X$_L$i (A1155463: 0.5 μM). DBP Assessed at 48 hours, peptide concentrations: ABT-199: 1 μM, MS1: 10 μM, HRK: 100 μM. Apoptosis (Annexin V/PI) assessed at five days. All box plots: mean, hinges at 25th and 75th percentiles, ± min to max.

or the combination of TMZ and IR (IR/TMZ) is often provided to GBM patients[33]. Like IR treatment, TMZ suppressed GBM cell proliferation but had a marginal impact on apoptosis (Supplementary Fig. 2D). Moreover, DBP revealed TMZ induced a specific reliance on BCL-X$_L$ for survival in a p53-dependent fashion (Fig. 2I, Supplementary Fig. 2E, F). Similarly, the IR and TMZ combination elicited minimal apoptosis and a sole dependency on BCL-X$_L$ for survival exclusively in p53 WT GBM cells (Fig. 2J, Supplementary Fig. 2D). These results show that IR, TMZ or the combination of IR/TMZ therapy rewires the apoptotic machinery in p53 WT GBM, consequently triggering a targetable vulnerability on BCL-X$_L$ for tumour cell survival.

## p53-mediated induction of PUMA ablates the MCL-1 block following standard-of-care therapy

p53 can modulate MCL-1 by inducing the expression of the sensitizer NOXA, which then binds to and neutralizes MCL-1[38]. Alternatively, p53 can suppress the level of MCL-1 protein by promoting MCL-1 ubiquitination and degradation[39,40]. However, we observed minimal changes in the protein expression of either NOXA or MCL-1 in p53 WT gliomaspheres following IR or TMZ treatment (Fig. 3A, Supplementary Fig. 3A). A broader investigation into the levels of BCL-2 family proteins revealed that the pro-apoptotic BH3 protein, PUMA, was the only BCL-2 family protein with significant changes in expression in p53 WT gliomaspheres

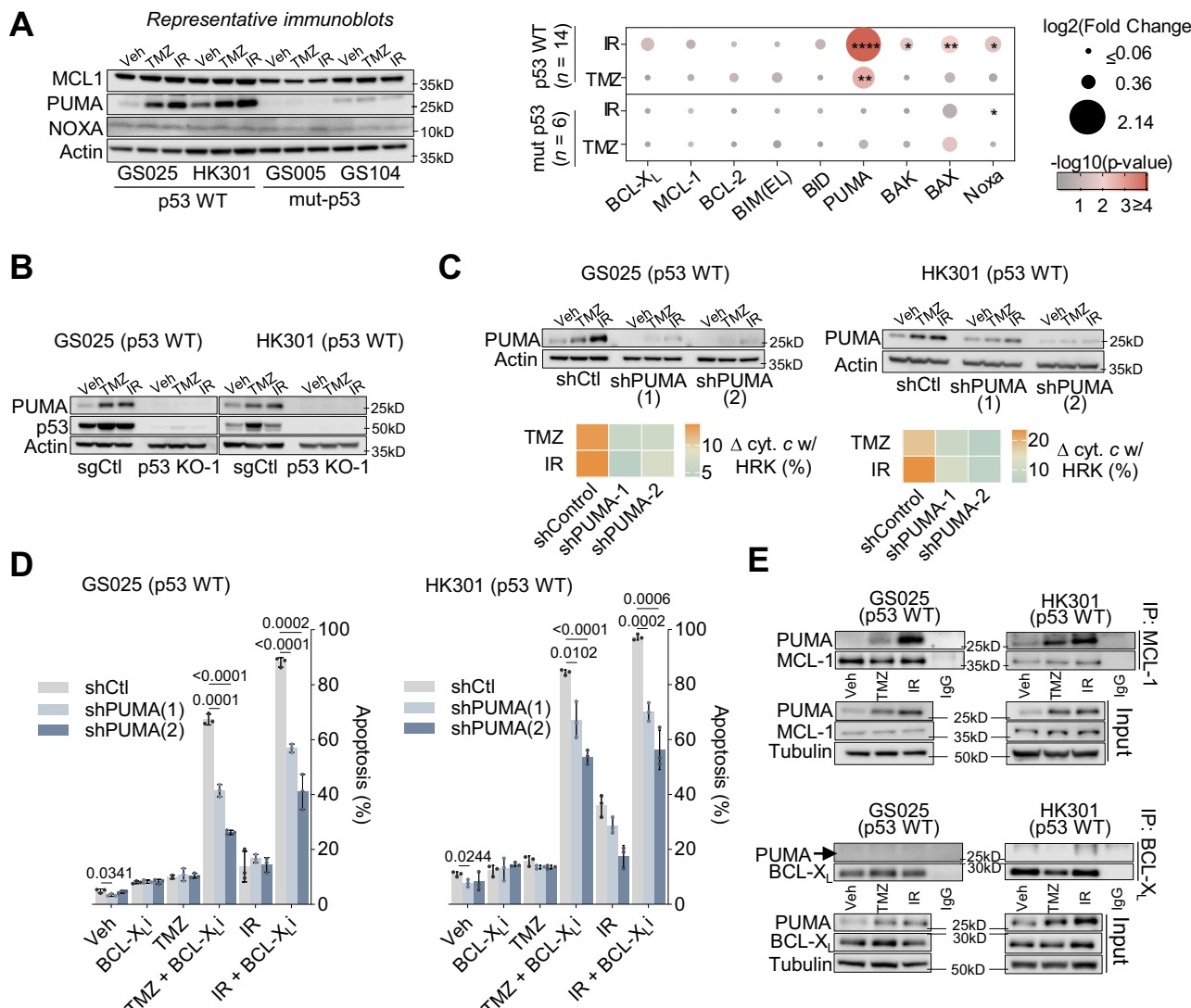

**Fig. 3 | p53-mediated induction of PUMA ablates the MCL-1 block following IR or TMZ therapy. A** Immunoblots of MCL-1, PUMA and Noxa expression 48 hours post IR (5 Gy) or TMZ (50 µM) treatment, in 4 out the 20 gliomasphere lines assessed, all blots shown in Supplementary Fig. 3A. Dot plot of quantified immunoblots across p53 WT ($n = 14$) and mut-p53 ($n = 6$) gliomaspheres. One sample t test compares if p53 WT or mut-p53 changed relative to their vehicle gliomaspheres for all proteins assessed. Colors represent -log10 transformed p-values and dot sizes represent log2 transformed fold changes. log2 fold changes less than or equal 0.06 were set as the minimum for point sizes, and maximum -log10 p-value was set to 4 (p-value = 0.0001). For full immunoblots of all lines see Supplementary Fig. 3. These results were independently repeated. **B** PUMA expression in p53 KO models in GS025 and HK301, 48 hours post TMZ and IR treatment. **C** Immunoblots of GS025-shControl, shPUMA-1, and shPUMA-2, 48 hours after TMZ (50 µM) or IR (5 Gy) treatment. DBP of GS025-shControl, shPUMA-1, and shPUMA-2, 48 hours after TMZ (50 µM) or IR (5 Gy) treatment, shows change in precent cytochrome *c* release with HRK (100 µM) (mean ± s.d., unpaired t test with Welch correction, $n = 2$ biological replicates). **D** Apoptosis (Annexin V/PI +) was evaluated in GS025-shControl, shPUMA-1, and shPUMA-2, 5 days after treatment with A1155463 (BCL-X$_L$: 0.5 µM), TMZ (50 µM), IR (5 Gy), TMZ + A1155463, or IR + A1155463 (mean ± s.d., unpaired t test with Welch correction, $n = 3$ biological replicates). **E** Co-immunoprecipitation of p53 WT gliomaspheres of BCL-X$_L$ or MCL-1 with PUMA following TMZ or IR. Gliomaspheres were treated for 48 hours with TMZ (50 µM) or IR (5 Gy). Signal quantification of PUMA determined relative to signal of MCL-1 or BCL-X$_L$ that was pulled down. These results were independently repeated. $p > 0.05$ = ns; $p < 0.05$ = *$p < 0.01$ = **$p < 0.001$ = ***$p < 0.0001$ = ****.

in response to IR or TMZ (Fig. 3A, Supplementary Fig. 3B). p53 KO confirmed that treatment-mediated PUMA protein upregulation was p53-dependent (Fig. 3B, Supplementary Fig. 3C). To test whether PUMA is responsible for negating the MCL-1 block, we used shRNA to reduce PUMA expression in two p53 WT gliomaspheres (GS025 and HK301) (Fig. 3C). In both gliomasphere lines, the loss of PUMA prevented the treatment-induced sole dependency on BCL-X$_L$ and mitigated cell death when challenged with IR or TMZ together with BCL-X$_L$i (Fig. 3C, D).

Although NOXA is perhaps the most recognized neutralizer of MCL-1, PUMA also has the capacity to bind with high affinity to MCL-1[41]. In p53 WT gliomaspheres, IR or TMZ-mediated PUMA induction coincided with an increase in PUMA-MCL-1 complexes (Fig. 3E). While we

also observed a modest induction in PUMA binding to BCL-X$_L$ with IR or TMZ treatment, this association was less than what was observed between PUMA and MCL-1 (Fig. 3F). Together, these results support that, following IR or TMZ treatment, p53 promotes the cellular accumulation of PUMA that subsequently binds to and inactivates MCL-1, and thereby creates an exclusive dependence on BCL-X$_L$ for GBM survival (Supplementary Fig. 3B).

## *TP53* genetic status cannot fully identify apoptotic vulnerable GBM

Our initial mechanistic studies suggested that *TP53* mutation status may be used as a genetic biomarker to predict susceptibility to

intrinsic apoptosis with either IR or TMZ combined with BCL-X$_L$ inhibition. To corroborate this observation, we extended these combination efficacy studies in an additional 18 gliomaspheres (total $n = 26$). We treated each gliomasphere line with either IR or TMZ plus BCL-X$_L$i and assessed apoptosis relative to IR or TMZ alone. Gliomaspheres with increased cell death above the mean of the treatment group were classified as sensitive to the respective combination (Fig. 4A). Curiously, while all mut-p53 GBMs ($n = 9$) were insensitive to the treatment combinations, we observed a substantial proportion of p53 WT gliomaspheres (IR + BCL-X$_L$i: $n = 7/17$; TMZ + BCL-X$_L$i: $n = 8/17$) were also non-responsive to the combination strategy (Fig. 4A). Thus, *TP53* mutational status alone cannot predict responses to targeting the glioma intrinsic apoptotic machinery with either IR or TMZ in combination with BCL-X$_L$ inhibition.

We next asked whether other molecular alterations within the p53 signaling pathway might affect the tumour response to the IR or TMZ + BCL-X$_L$i therapeutic combinations. MDM2 negatively regulates p53 signaling[42], and its genetic locus can be amplified in patients with wild type p53 gliomas[43]. Indeed, *MDM2* amplified gliomaspheres ($n = 2$), which are p53 WT, were insensitive to IR or TMZ + BCL-X$_L$i (Fig. 4A).

Other known molecular biomarkers in GBM include the DNA repair protein O-6-methylguanine-DNA methyltransferase (MGMT), which repairs the DNA damage induced by TMZ treatment[44]. Consequently, tumours with an unmethylated MGMT promoter and with corresponding high MGMT RNA expression (MGMT positive) are generally insensitive to TMZ[44]. We observed that most p53 WT gliomaspheres that were also MGMT positive were non-responsive to TMZ and BCL-X$_L$i ($n = 8/10$) (Fig. 4A). We confirmed the requirement for suppressed levels and activity of MGMT for TMZ and BCL-X$_L$i sensitivity both genetically and pharmacologically (Supplementary Fig. 4A, B).

However, while incorporation of both *MDM2* amplification and MGMT expression status improved on the classification of p53 WT gliomaspheres insensitive to the combination strategy (Fig. 4A), approximately 30% of the remaining intact p53 signaling glioma cells were nonresponsive to treatment (IR + BCL-X$_L$i: $n = 5/15$; TMZ + BCL-X$_L$i: $n = 3/10$). Consequently, enrichment for gliomas with the compilation of molecular biomarkers (*TP53* WT, *MDM2* WT, and MGMT negative) was still unable to fully distinguish between therapeutic responders and non-responders (Fig. 4B). Our findings suggested that this set of mechanistically defined molecular biomarkers could not fully identify the gliomas sensitive to IR or TMZ + BCL-X$_L$ inhibition.

### Functional assessment of the apoptotic 'primed state' alone cannot fully predict response to apoptotic targeting

A tumour cell's apoptotic potential (i.e., mitochondria priming) indicates its proximity to the apoptotic threshold. Accordingly, a functional assessment of a tumour's mitochondria priming, via BH3 profiling, prior to treatment can predict therapy-induced intrinsic apoptosis[32]. To explore whether mitochondrial priming status can identify if GBMs are responsive or non-responsive to IR or TMZ with BCL-X$_L$i treatment, we measured the apoptotic potential of our panel of gliomaspheres ($n = 26$) by performing BH3 profiling with increasing concentrations of the pro-apoptotic BIM BH3 peptide (equal affinity to all anti-apoptotic proteins)[41] (Fig. 4C). We calculated the area under the curve of cytochrome *c* release to create a metric to represent the primed state of each gliomasphere tested (BIM$^{AUC}$) (Supplementary Fig. 4C). This analysis revealed that gliomaspheres exhibit diversity in mitochondria priming.

Next, we asked whether the differences in mitochondria priming among our gliomasphere cohort could stratify responders and non-responders to IR or TMZ + BCL-X$_L$i treatment. Using receiver operating characteristic (ROC) curves to establish a binary classification of

'highly primed' and 'lowly primed' gliomaspheres (Supplementary Fig. 4D), we observed that while all lowly primed gliomaspheres were insensitive to the IR or TMZ + BCL-X$_L$i combination (Fig. 4D), there remained a considerable number of highly primed gliomaspheres that did not respond to treatment (IR + BCL-X$_L$i: $n = 5/15$; TMZ + BCL-X$_L$i: $n = 6/15$) (Fig. 4D). Therefore as was observed with defined molecular biomarkers, functional BH3 profiling alone could not fully stratify GBMs responsive to this combination therapy approach.

### Selection for intact p53 signaling molecular status increases the association between primed apoptotic state and therapeutic response

To further investigate the highly primed GBMs that were insensitive to IR or TMZ + BCL-X$_L$i, we performed univariate linear regressions to test the association between treatment-induced apoptosis and functional BH3 profiling (Fig. 4E, F). We found highly primed gliomaspheres having a low response to IR or TMZ + BCL-X$_L$i contained a molecular alteration in the p53 signaling pathway (e.g., GS116 and GS121: mut-p53). Therefore, we hypothesised that enriching the population for gliomaspheres with intact p53 signaling (e.g., *TP53* WT, *MDM2* WT and MGMT negative) would enhance the association between apoptotic potential and drug sensitivity. Accordingly, adding these molecular feature requirements substantially strengthened the correlation between apoptotic priming and response to the combined therapeutics (IR + BCL-X$_L$i: $R^2 = 0.41 \rightarrow 0.58$; TMZ + BCL-X$_L$i: $R^2 = 0.35 \rightarrow 0.82$) (Fig. 4E, F). Consequently, enriching for both p53 signaling status and apoptotic potential could now identify >90% (IR + BCL-X$_L$i: 91%; TMZ + BCL-X$_L$i: 100%) of the gliomaspheres that responded to IR or TMZ + BCL-X$_L$i. By contrast, using p53 signaling mutational status or BH3 functional profiling alone identified ~70% of the gliomaspheres responsive to the therapeutic combinations (Fig. 4E, F).

### A machine-learning model identifies a composite molecular and functional predictive biomarker for targeting GBM apoptosis

Our results suggest that specific molecular and functional characteristics are required for GBM cell death in response to targeting the apoptotic machinery. To explore whether an integrated biomarker consisting of both genomic and functional features could best predict GBM tumours sensitive to this apoptotic targeting approach, we sought to develop a machine-learning model using these diverse datasets. We trained and compared the performance of two predictive models: the first trained on a combination of previously implicated molecular features (*TP53* mutational status, *MDM2* amplification status, MGMT status) as well as functional (BIM$^{AUC}$) components to create an Integrated Molecular and Functional (IMF) feature set ($p = 15$). The second model trained on Global Molecular (GM) features including gene expression ($p = 18,909$ genes) and genomic alterations (copy number alterations: $p = 19,023$; somatic mutations: $p = 213$) (Fig. 5A, Supplementary Table 2). Model predictions were based on elastic net regressions subject to nested cross-validation (see Methods).

The IMF model frequently selected features with interactions between p53 signaling genetic status (as well as MGMT for TMZ) and apoptotic potential (Supplementary Fig. 5A) and predicted GBM sensitivity values for IR + BCL-X$_L$i ($r^2 = 0.74$, RMSE (Root Mean Square Error) = 15.36) and TMZ + BCL-X$_L$i ($r^2 = 0.64$, RMSE = 14.69) (Fig. 5B, Supplementary Fig. 6A). In the GM model, almost all features selected were transcriptomic, with many of the genes relating to pathways linked with apoptosis, DNA damage repair, and p53 signaling (Supplementary Fig. 5B, C). Relative to the IMF model, the GM model was less effective at predicting apoptosis in response to IR + BCL-X$_L$i ($r^2 = 0.0003$, RMSE = 37.69) and TMZ + BCL-X$_L$i ($r^2 = 0.15$, RMSE = 24.40) (Fig. 5B, Supplementary Fig. 6A). Next, we pooled the IMF and GM features and ranked their predictive importance using a Least Absolute Shrinkage and Selection Operator (LASSO) regression approach (see Methods). We found the top

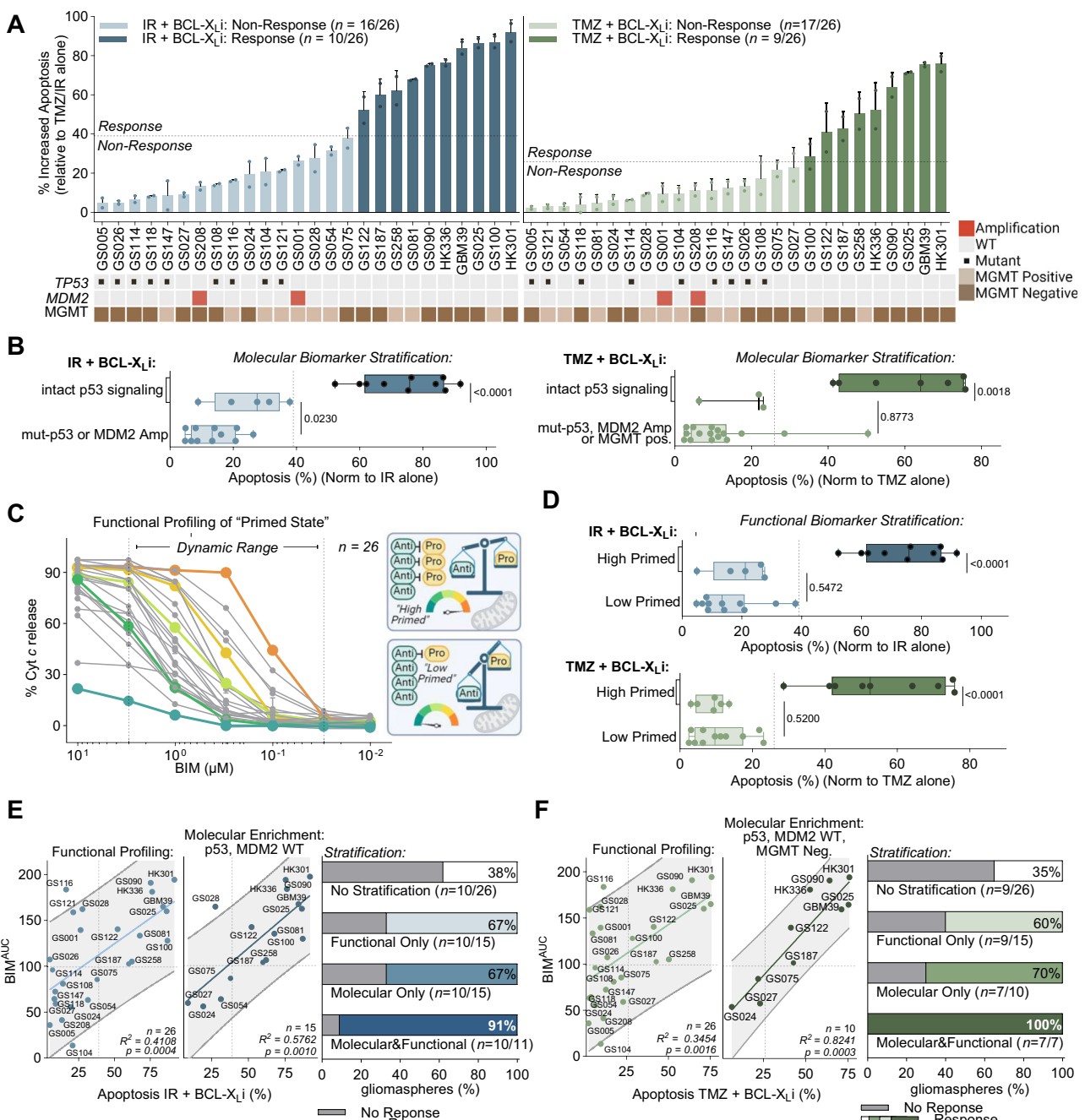

**Fig. 4 | p53 genetic status alone cannot predict response to TMZ/IR in combination with BCL-X$_L$i. A** Apoptosis (Annexin V/PI +) of gliomaspheres ($n$ = 26) 5 days post IR (5 Gy) or TMZ (50 μM) and BCL-X$_L$i (A1155463: 0.5 μM) treatment normalized to either IR or TMZ alone (mean ± s.d., $n$ = 2 independent replicates). Response cutoff determined by taking the mean. Heat map below displays mutations or copy number alterations in *TP53* or *MDM2*, respectively. MGMT status is determined using MGMT methylation and expression, see methods. **B** Grouped analysis of apoptosis visualized with box plots (mean, hinges at 25th and 75th percentiles, ± min to max) of all molecular biomarker positive vs negative gliomaspheres, sub-divided by response ($n$ = 26), (two-tailed, unpaired t test). Darker coloring signifies responders identified in 4 A. **C** Basal BH3 profiling of gliomaspheres ($n$ = 26) pre-formed with a titration of the BIM peptide (0 μM, 0.01 μM, 0.03 μM, 0.1 μM, 0.3 μM,

1 μM, 3 μM, 10 μM). Data points in the dynamic range (0.03 μM − 3 μM BIM, defined by peptides with the greatest range in responses) used to calculate area the AUC. Diagram depicts simplified examples of high and low primed gliomaspheres. **D** Grouped analysis of apoptosis visualized with box plots (mean, hinges at 25th and 75th percentiles, ± min to max) of all functional biomarker positive vs negative gliomaspheres, sub-divided by response ($n$ = 26), (two-tailed, unpaired t test). Darker coloring signifies responders identified in 4 A. Correlations of BIM$^{AUC}$ with a simple linear regression between cell death induced by IR (**E**) or TMZ (**F**) and BCL-X$_L$i, grey band represents with 95% confidence interval. Line represents simple linear regression, used to calculate p-value and r-squared. Summary bar plots describing the percent of correctly identified gliomaspheres sensitive to with the different biomarker stratification.

feature for IR + BCL-X$_L$i was BIM$^{AUC}$*TP53*MDM2 and for TMZ + BCL-X$_L$i was BIM$^{AUC}$*TP53*MDM2*MGMT (Supplementary Fig. 6C).

To confirm these results, we evaluated the prospective ability of the IMF and GM models to predict sensitivity among an independent

verification cohort of patient-derived gliomaspheres ($n$ = 12 gliomasphere lines, Supplementary Table 3) (Fig. 5C, Supplementary Fig. 6A, B, Supplementary Table 4). Like the training cohort, the IMF model (IR + BCL-X$_L$i: $r^2$ = 0.66, RMSE = 17.45; TMZ + BCL-X$_L$i: $r^2$ = 0.72,

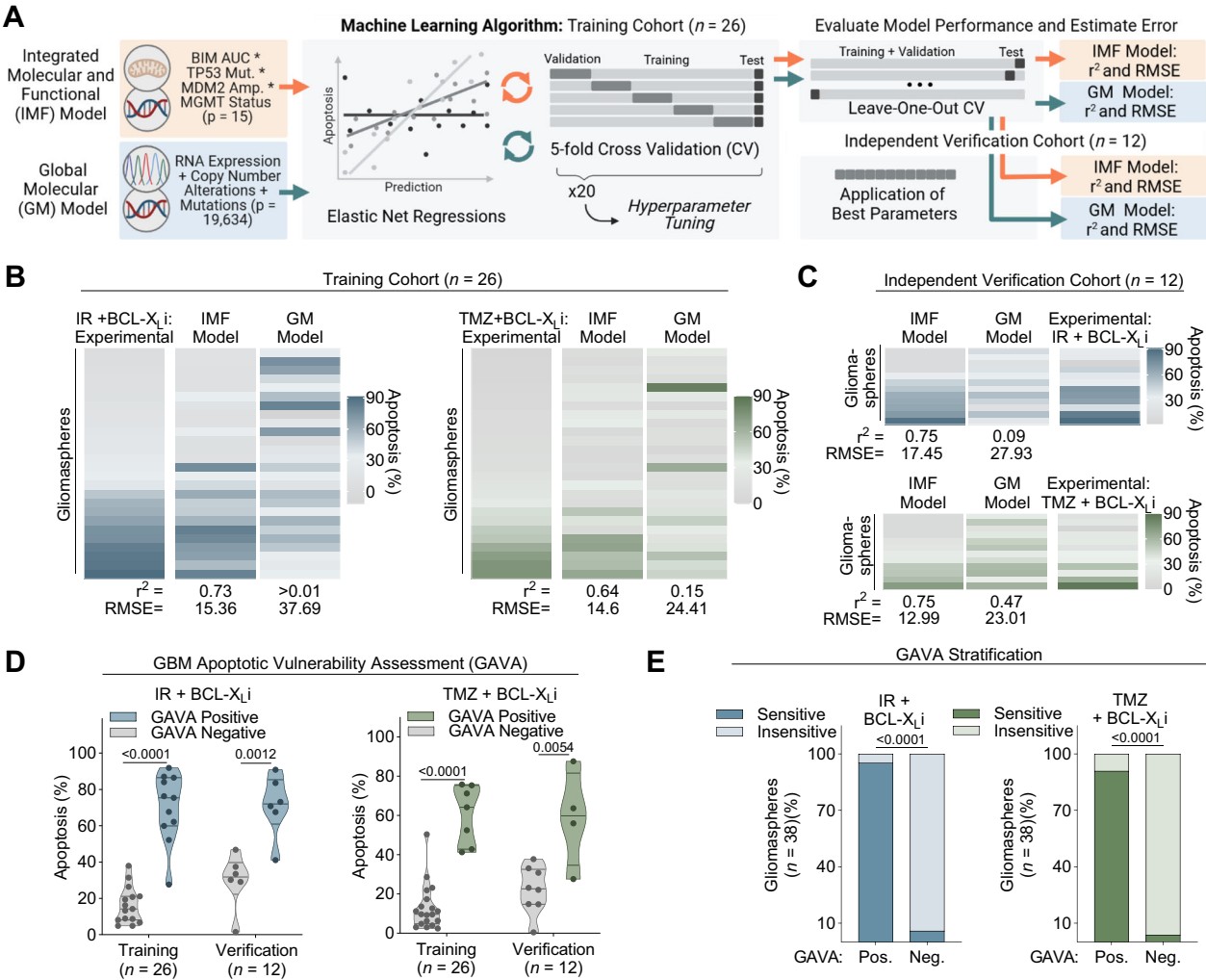

**Fig. 5 | Development and predictability of integrated genomic and functional biomarker. A** Schematic outlining the machine learning algorithm developed to compare the Integrated Molecular and Functional (IMF) model with the Global Molecular (GM) model through elastic net regression subject to cross validation (methods section). **B** Training Cohort ($n = 26$ gliomaspheres): Heatmaps display experimental (IR (5 Gy) or TMZ (50 μM) + BCL-X$_L$i(0.5 μM) and predicted values for either IMF or GM models, for each gliomasphere. R$^2$ and RSME, used to estimate error of the model calculated for each model (methods section). **C** Independent Verification Cohort ($n = 12$ gliomaspheres): Heatmaps display experimental (IR or TMZ + BCL-X$_L$i) and predicted values for either IMF or GM models, for each gliomasphere. R$^2$ and RSME, used to estimate error of the model calculated for each model (methods section). **D** Grouped analysis of all GAVA positive vs GAVA negative gliomaspheres, for both the training and verification cohorts. IR (5 Gy) or TMZ (50 μM) + BCL-X$_L$i(0.5 μM) cell death plotted as violon plots (two-tailed, unpaired t test). **E** GAVA stratification applied to both training and verification gliomaspheres ($n = 38$) (Fischer's exact test).

RMSE = 12.99) provided better predictive capability than the GM model (IR + BCL-X$_L$i: $r^2 = 0.12$, RMSE = 27.93; TMZ + BCL-X$_L$i: $r^2 = 0.19$, RMSE = 23.01). Thus, even relative to global molecular datasets, integrated molecular profiling of the p53 signaling pathway with functional profiling of apoptotic potential best predicts IR or TMZ + BCL-X$_L$i-induced apoptosis in heterogeneous patient-derived gliomaspheres.

Finally, we set out to define a simple model for stratifying GBM samples into those predicted as sensitive or insensitive to inducers of apoptosis based on the most informative features identified through machine learning. We applied the top-ranked IMF features (IR + BCL-X$_L$i = BIM$^{AUC}$*TP53*MDM2; TMZ + BCL-X$_L$i = BIM$^{AUC}$*TP53*MDM2*MGMT) (Supplementary Fig. 6C) as binary variables to score the gliomaspheres. We termed this scoring approach the GBM Apoptotic Vulnerability Assessment (GAVA), where gliomaspheres containing both the identified molecular and functional features were designated GAVA positive, and those lacking either of these defined molecular or functional features were designated as GAVA negative (see Methods).

GAVA-positive gliomaspheres from both training and verification cohorts were more sensitive to IR or TMZ + BCL-X$_L$i relative to gliomaspheres designated GAVA negative from both cohorts (Fig. 5D). Moreover, the GAVA biomarker had strong predictive power, achieving an AUROC of 0.99 and 0.97 for IR + BCL-X$_L$i in the training and verification cohorts, respectively, and an AUROC of 0.99 and 0.91 for TMZ + BCL-X$_L$i in the training and verification cohorts, respectively (Supplementary Fig. 7A). Finally, across all gliomaspheres ($n = 38$ in total), 95% (IR + BCL-X$_L$i) and 91% (TMZ + BCL-X$_L$i) of gliomaspheres designated as GAVA positive were sensitive to apoptotic targeting therapy, while 94% (IR + BCL-X$_L$i) and 96% (TMZ + BCL-X$_L$i) of GAVA-negative gliomaspheres were insensitive to these therapeutic combinations (Fig. 5E). Together, these data indicate GAVA may serve as an integrated molecular and functional predictive biomarker for this therapeutic approach in GBM.

**The antibody-drug conjugate ABBV-155 targets BCL-X$_L$ in glioma**
Our results support the conclusion that gliomas with intact p53 signaling and high mitochondria priming are susceptible to the

drug combination of IR or TMZ and a BCL-X$_L$ inhibitor. This suggests a potential therapeutic strategy and biomarker for targeting glioma intrinsic apoptosis. However, BH3 mimetics that block BCL-X$_L$ (e.g., Navitoclax) have a poor therapeutic window in patients, in part due to on-target, dose-limiting toxicity in platelets, which depend on BCL-X$_L$ for survival[45]. Mirzotamab Cletuzoclax (ABBV-155) is a first-in-class antibody-drug conjugate (ADC) that targets the cell surface protein B7-H3 (CD276) and includes a specific and potent BCL-X$_L$ inhibitor warhead[46,47]. B7-H3 is highly expressed in glioma, as well as other solid tumours, and is a compelling target for ADCs as well as cell-based therapies given its high tumour-to-normal tissue expression[48,49]. Notably, early clinical data indicate ABBV-155 is safe in humans with a maximum tolerated dose (MTD) not reached and, unlike previous inhibitors of BCL-X$_L$, does not impact platelets to a significant degree (i.e., no thrombocytopenia)[46].

To first evaluate the protein expression of B7-H3 in GBM tumours relative to normal brain specimens, we performed immunohistochemistry of B7-H3 on a tissue microarray of GBM samples ($n = 34$) and normal brain biopsies ($n = 5$). In agreement with previous findings[49], we observed high and diffuse B7-H3 protein expression on GBM cells, whereas expression in normal brain specimens was largely absent (Fig. 6A, Supplementary Fig. S8A). We corroborated these findings by immunoblotting for B7-H3 protein expression in autopsy samples and gliomaspheres lines (Supplementary Fig. S8B, C).

Next, to assess whether ABBV-155 could selectively abolish the BCL-X$_L$ block in our gliomaspheres, we performed DBP across six gliomasphere lines ($n = 4$ p53 WT, $n = 2$ mut-p53). ABBV-155 created a single dependency on MCL-1 to prevent MOMP in all gliomaspheres (Fig. 6B), and consequently promoted apoptosis when combined with MCL-1 inhibition (Supplementary Fig. 8D). Therefore, ABBV-155 can selectively remove the BCL-X$_L$ block regardless of gliomasphere *TP53* genetic status.

To test if ABBV-155 promotes GBM cell death when combined with IR and, if so, whether GAVA could predict apoptotic sensitivity, we treated a mixed population of GAVA-positive and GAVA-negative gliomaspheres ($n = 7$) with IR in combination with increasing concentrations of ABBV-155. We observed that IR + ABBV-155 induced apoptosis in the GAVA-positive gliomaspheres, GS025 and GBM39, with cell death observed as low as 0.001 μg/ml of ABBV-155 when combined with IR (Fig. 6C). By contrast, all GAVA negative gliomaspheres ($n = 5$) were insensitive to IR + ABBV-155 (Fig. 6D–F). Similarly, TMZ + ABBV-155 triggered synergistic apoptosis specifically in GAVA-positive gliomaspheres (Supplementary Figure 8E–G). Thus, GAVA-predicted apoptotic sensitivity in response to ABBV-155 combined with standard-of-care therapies in GBM cells (Fig. 6F, Supplementary Fig. 8H).

Importantly, a non-targeting control (NTC) ADC with the same potent BCL-X$_L$ inhibitor warhead as ABBV-155 was incapable of inducing cell death either alone or in combination with IR or TMZ in the highly primed p53 WT line, GS025 (Fig. 6G, Supplementary Fig. 8I). These data suggest that the synergistic lethality of ABBV-155 in GBM cells, when combined with IR or TMZ, is specific for B7-H3 expression.

To extend our characterisation of ABBV-155 in GBM, we asked whether ABBV-155 could ablate the BCL-X$_L$ block in vivo. Accordingly, we established both a p53 WT (PDX039) and a mut-p53 (PDX147) patient-derived orthotopic GBM xenograft in the brains of immunocompromised mice. Once tumours were established, mice were treated with one dose of ABBV-155 (10 mg/kg) or vehicle. Tumours were harvested 72 hours later, and tumour cells were purified and subjected to DBP. Consistent with our in vitro results, we found that ABBV-155 caused a single dependency on MCL-1 in both xenograft models, supporting the conclusion that ABBV-155 can ablate the BCL-X$_L$ block in orthotopic glioma xenografts (Fig. 6).

## Integration of genomic and functional profiling predicts in vivo tumour response to IR + ABBV-155

We next explored if ABBV-155 sensitizes GBM tumours to DNA-damaging therapy in vivo and whether the GAVA biomarker could prospectively determine the tumours most vulnerable to this combination. Accordingly, we established orthotopic xenograft models ($n = 6$) and, once engrafted, harvested the tumours from mice and purified the tumour cells for genomic characterisation and functional BH3 profiling of the primed state (Fig. 7A). Our analysis revealed PDX025 and PDX039 were both *TP53* and *MDM2* WT as well as primed for apoptosis (Fig. 7A). These samples were classified as GAVA positive; therefore, we predicted these two glioma tumour xenografts would be sensitive to IR + ABBV-155 treatment. Conversely, PDX054 and PDX027 were *TP53* WT, *MDM2* WT but lowly primed, and PDX147 and PDX005 were both mut-p53 (Fig. 7A). These tumours were collectively classified as GAVA negative since they lacked both the requisite molecular and functional characteristics proposed for therapeutic response to IR in combination with ABBV-155. We hypothesised these orthotopic xenografts would not respond to such combination therapies.

To test these predictions, we established a separate cohort of intracranial tumours from these six PDXs. Following intracranial inoculation, mice were monitored for exponential tumour growth by secreted gaussia luciferase[50]. Once aggressive tumour growth was achieved, mice were randomised and treated with 10 mg/kg ABBV-155 intraperitoneally (i.p.) weekly for three weeks, 10 Gy IR weekly for two weeks, or the combination of ABBV-155 and IR. ABBV-155 did not impact tumour growth in five out of six models, with only modest changes in PDX025 (relative to vehicle treated mice) and did not extend survival in all PDX models tested. IR was capable of significantly inhibiting tumour growth − albeit largely tumour static responses − and prolonging survival in both p53 WT and mut-p53 models. By contrast, ABBV-155 + IR significantly reduced tumour size and prolonged mouse survival relative to single agent controls; however, this combined therapeutic efficacy was confined to PDXs that were GAVA positive (PDX025 and PDX039; Fig. 7B–D). None of the GAVA-negative PDXs showed tumour regressions or augmented survival with ABBV-155 in combination with IR therapy. Thus, we conclude that the combination of DNA-damaging therapy and ABBV-155 can have an anti-tumour impact in aggressive glioma PDX models that are both p53 WT and primed for apoptosis.

Our in vitro and in vivo findings demonstrate that integrated molecular and functional profiling by GAVA can robustly predict responses to this approach to target GBM intrinsic apoptosis. To consider the translational potential of GAVA, we analyzed the p53 genomic status as well as the apoptotic priming of 21 WHO grade 4 glioma purified patient tumour samples (from Fig. 1). 62% percent of our patient sample cohort had intact p53 signaling genetic features (*TP53* WT, *MDM2* WT), which is concordant with the 67% of patients having these features as reported by TCGA[31]. BH3 profiling revealed that 50% of our patient samples were highly primed, with an even distribution of primed samples among p53 WT and altered tumours. Accordingly, 31% of our patient sample cohort were GAVA positive (Fig. 7E). These results support that integrated molecular and functional profiling can be performed on freshly isolated glioma patient samples, revealing that a considerable number of glioma patients may be potential candidates for this proposed predictive biomarker and therapeutic combination approach assaying and targeting intrinsic apoptosis.

## Discussion

Precision medicine for cancer is currently largely dependent on static genomic biomarkers. However, genetic features cannot always predict tumour cell function, particularly in response to drug perturbations[51].

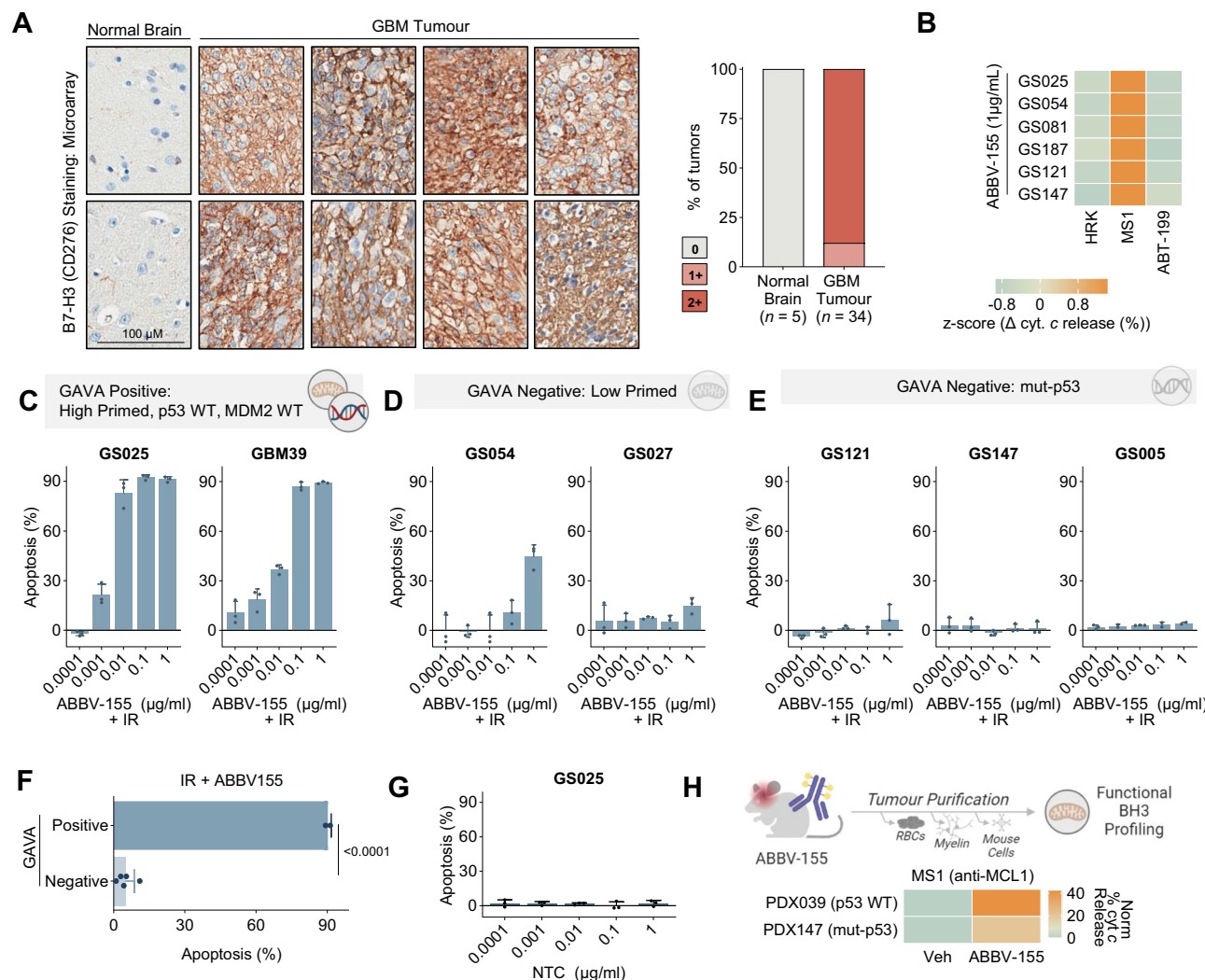

**Fig. 6 | ABBV-155 (Clezutoclax-B7-H3 ADC) targets BCL-X$_L$ in GBM.**
**A** Immunohistochemistry staining for B7-H3 (CD276) in GBM ($n = 34$) and normal brain ($n = 5$) microarray. Representative images of two out of five normal brains and 8 out of 33 GBMs. Staining was scored as, 0 (negative in glial cells), 1+ (weakly positive) and 2+ (strongly positive). **B** DBP 72 hours post ABBV-155 (1 µg/mL) treatment of gliomaspheres ($n = 6$). Data plotted as z-score across sample showing change with treatment with single peptides relative to the double block (ΔHRK + MS1-ΔHRK/ΔMS1/ΔABT-199). Peptide concentrations are as follows: ABT-199: 1 µM, MS1: 10 µM, HRK: 100 µM. **C** Apoptosis (Annexin V/PI +) of GAVA positive gliomaspheres (p53 and MDM2 WT and high primed), GS025 and GBM39, 5 days post IR (5 Gy) and ABBV-155 titration. Concentration range: 0.0001 µg/mL, 0.001 µg/mL, 0.01 µg/mL, 0.1 µg/mL, 1 µg/mL. Data are normalized to R alone ($n = 3$ biological replicates). **D** Apoptosis (Annexin V/PI +) of GAVA negative gliomaspheres (low primed), GS054 and GS027, 5 days post IR (5 Gy) and ABBV-155 titration.

Concentration range: 0.0001 µg/mL, 0.001 µg/mL, 0.01 µg/mL, 0.1 µg/mL, 1 µg/mL. Data are normalized to R alone ($n = 3$ biological replicates). **E** Apoptosis (Annexin V/PI +) of GAVA negative gliomaspheres (mut-p53), GS121, GS147 and GS005, 5 days post IR (5 Gy) and ABBV-155 titration. Concentration range: 0.0001 µg/mL, 0.001 µg/mL, 0.01 µg/mL, 0.1 µg/mL, 1 µg/mL. Data are normalized to R alone ($n = 3$ biological replicates). **F** Grouped analysis by GAVA status of all IR + 0.1 µg/mL ABBV-155 cell death data (mean ± s.d., two-tailed, $n = 7$ gliomaspheres). **G** Apoptosis (Annexin V/PI +) of p53 WT gliomaspheres, GS025, 5 days post IR (5 Gy) and Non-Targeting Control (NTC). Concentration range: 0.0001 µg/mL, 0.001 µg/mL, 0.01 µg/mL, 0.1 µg/mL, 1 µg/mL. Data are normalized to IR alone ($n = 3$ biological replicates). **H** PDX039 (p53WT) and PDX147 (mut-p53) were treated with ABBV-155 (10 mg/kg, i.p., *qw*). 10 days later purified tumour cells were used to perform ex-vivo DBP. Data indicate a change in cytochrome *c* release between ABBV-155 treated and vehicle untreated mice with MS1(MCL1 - 10 µM) ($n = 2$ xenografts).

In this study, we demonstrate a precision medicine approach in which we integrate genomic and functional measurements to determine GBM tumour responses to therapy targeting intrinsic apoptosis. Specifically, we show that standard-of-care DNA-damaging therapy (IR or TMZ) shifts the composition of the GBM intrinsic apoptotic machinery with genetically intact p53 signaling (i.e., *TP53* WT, *MDM2* WT). Moreover, while synergistic apoptosis with combined IR or TMZ and BCL-X$_L$ inhibition was exclusive to p53 WT tumours, functional BH3 profiling of apoptotic potential was also required to predict response to the drug combination, both in vitro and in vivo. This finding enabled us to use machine learning to establish an integrated molecular and

functional predictive biomarker – which we term GAVA – for targeting glioma-intrinsic apoptosis.

Our findings may have important therapeutic implications for treating malignant glioma, a cancer notoriously refractory to cell death. We show that the anti-apoptotic proteins BCL-X$_L$ and MCL-1 both protect against cell death across all glioma tumours, regardless of genetic features. This observation suggests a dual apoptotic barrier exists for glioma that must be overcome for a glioma cell to execute apoptosis. We demonstrate that DNA-damaging therapies (i.e., the standard of care for GBM) rapidly ablate the MCL-1 block via p53-dependent induction of PUMA, which subsequently binds and

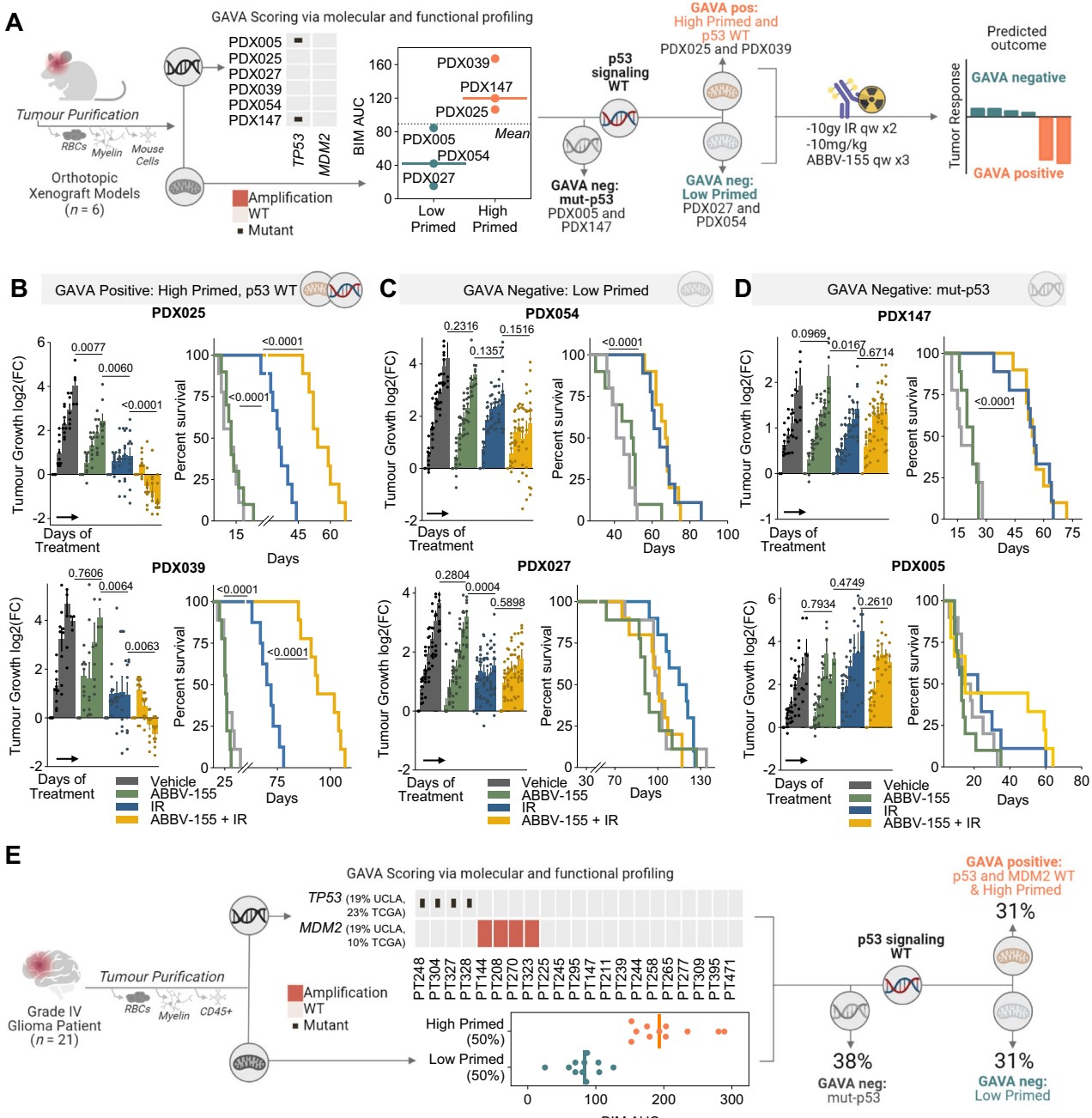

**Fig. 7 | Integration of genomic and functional profiling predicts tumour response to IR + ABBV-155. A** Workflow describing GAVA application of GBM orthotopic xenografts ($n = 6$) to predict response to IR and ABBV-155. Scatter dot plots show the $BIM^{AUC}$ from ex vivo basal BH3 profiling of indicated untreated orthotopic xenografts. Designation as high and low primed determined by the median of the ex vivo samples. The following schematic details experiment work-flow and treatment schedule, followed by predicted response by GAVA status. **B, C** Assessment of changes in GAVA positive (**B**), GAVA Negative (low primed) (**C**), and GAVA Negative (p53 mutant) (**D**) PDX tumour growth of mice treated with vehicle, ABBV-155 (10 mg/kg, i.p. qw x 3), IR (10 Gy, qw x 2), or the combination of ABBV-155 and IR. Fold change in tumour burden relative to size at enrollment displayed for all models (mean ± SEM, $n = 10$ SDX025, SDX054, SDX027, SDX147, SDX005, $n = 9$ SDX-GBM39). Grouped comparisons made were made with two-tailed, unpaired t tests using data sets from the last measurements. Percentage survival of orthotopic xenografts after the indicated treatments. Comparisons made using Log-rank (Mantel-Cox) test. **E** Workflow describing GAVA application on grade 4 patient samples ($n = 21$). Heat map shows mutations and amplifications for TP53 and MDM2. Scatter dot plots show the $BIM^{AUC}$ from the ex-vivo basal BH3 profiling of patient tumours. Designation as high and low primed determined using the median of the patient tumour samples.

sequesters MCL-1, creating a sole survival dependency on BCL-X$_L$ in GBM lacking genetic alterations in the p53 signaling pathway. Interestingly, other work has also identified BCL-X$_L$ dependencies in GBM cells after drug perturbations[52], including a BCL-X$_L$ dependency after drug-induced senescence[37]. How the MCL-1 block is modulated in these contexts, and the impact of genotype and apoptotic primed state on these reported apoptotic block dependencies is not clear.

Moreover, in *IDH* mutant gliomas the production of the oncometabolite, 2-HG, may reduce MCL-1 protein stability, which could also elicit BCL-X$_L$ dependencies[53]. These findings, together with what is described here, highlight the need to assess how other genetic alterations or therapeutic perturbations shift the dependencies on the MCL-1 and BCL-X$_L$ apoptotic blocks in various malignant brain tumours.

The exclusive dependence on BCL-$X_L$ created by standard-of-care therapy in p53 WT GBM that are highly primed (31%) suggest BCL-$X_L$ is an attractive target for many patients with GBM. While the dual BCL-$X_L$ and BCL-2 inhibitor, Navitoclax, has been evaluated clinically for non-CNS malignancies[54], it is dose-limited by the "on-target, off-tumour" toxicity in platelets (thrombocytopenia). Moreover, recent evidence indicates that selective inhibition of BCL-$X_L$ can trigger cardiovascular toxicity[47]. ABBV-155 is a clinical antibody drug conjugate that targets B7-H3 — a highly expressed tumour protein — and delivers a potent warhead specifically targeting BCL-$X_L$ in tumours. Early clinical evaluations of ABBV-155 show a favorable toxicity profile and promising therapeutic potential in non-CNS tumours[46]. Here we provide the first evaluation of ABBV-155 for brain tumours and demonstrate its potential to selectively target tumour BCL-$X_L$ in mouse intracranial patient-derived GBM models. While concerns exist as to whether ADCs can accumulate to sufficient levels in the CNS of patients with brain tumours, clinical evidence supports the activity of other ADCs used in glioma (e.g., Depatuxizumab Mafodotin (ABT-414)[55] and other CNS malignancies (e.g., Trastuzumab-deruxtecan for CNS metastatic breast cancer)[56]. Thus, ADCs including ABBV-155 may have therapeutic potential for glioma patients, particularly when the blood brain barrier (BBB) shows considerable permeability[57]. Additional studies are needed to understand the requirements to achieve adequate CNS exposures of ABBV-155 and other ADCs under clinical consideration for primary brain tumours.

Functional precision medicine is showing clinical promise in guiding treatment decisions for cancer patients[14,22,58,59], yet most of the FPM studies for solid tumours have relied on ex vivo drug screening of patient samples[16,60]. Here we demonstrate in 21 patient glioma resections that both molecular and BH3 profiling can be performed on freshly isolated glioma patient samples, without requiring cell culture. Our analysis revealed gliomas have both significant molecular heterogeneity as well as various states of apoptotic potential. The molecular underpinnings of apoptotic priming among gliomas remains unknown. Moreover, it is unclear whether isolating glioma cells from surrounding non-malignant cells in the tumour microenvironment influences the glioma apoptotic machinery. Future investigations into these important gaps are warranted. Nevertheless, the results presented here show that apoptotic priming and genomic alterations can be assessed directly from brain tumour patient specimens, highlighting the importance and feasibility of making both measurements. This could pave a path for the clinical translation of using an integrated biomarker, such as GAVA, to stratify glioma patients for this therapeutic approach.

Taken together, our findings identify mechanisms of resistance to intrinsic apoptosis across malignant gliomas and demonstrate how they can be therapeutically targeted to synergistically induce cell death and enhance cancer therapy. We evaluate an ADC to target BCL-$X_L$ in combination with the standard of care, and show the effectiveness of this strategy across patient-derived model systems. Finally, we propose a precision medicine methodology, which relies on both molecular and functional tumour characteristics measured directly from patient samples, to robustly predict tumour responses.

## Methods
This study complied with all the relevant ethical regulations. All patient-derived tumour tissue was obtained through the UCLA Institutional Review Board (IRB) protocol 10-000655, after written informed consent was obtained from patients. All studies were in accordance with UCLA OARO protocol guidelines and in accordance with UCLA Animal Research Committee protocol guidelines. All mice experiments were approved by the Institutional Animal Care and Use Committees at UCLA.

### Patient-derived GBM tumours and gliomaspheres
Tumour resections were mechanically and enzymatically dissociated using the Miltenyi Biotec Human tumour dissociation kit (130-094-929) within six hours of surgery, followed by removal of red blood cells with ACK lysis buffer (Gibco, A10492-01). Next, antibody-conjugated magnetic beads were used to remove CD45+ cells (Miltenyi, 130-045-801) and myelinated cells (Miltenyi, 130-096-433) by preforming column-based filtrations. Primary GBM cells were established and cultured as gliomaspheres in media consisting of DMEM/F12 (Gibco, 11330032), B27 (Invitrogen, 12587010), penicillin–streptomycin (Invitrogen, 15140122), and GlutaMAX (Invitrogen, 35050061) supplemented with heparin (5 mg/mL, Sigma, H3149), EGF (20 ng/mL, Sigma, PHG0313), and FGF (20 ng/mL, Sigma, PHG0263). When passaged, gliomaspheres were dissociated to single cell suspensions with TrypLE (Thermo Fisher, 12605028). All cells were grown at 37 °C, 20% $O_2$, and 5% $CO_2$ and were routinely tested and confirmed negative for the presence of mycoplasma (MycoAlert, Lonza, LT07-318). Gliomaspheres were used at fewer than 15 passages, except for HK301, HK336 and GBM39, which were used between 15 and 30 passages. All cells were authenticated by short tandem repeat (STR) analysis.

### Mice
Female NOD scid gamma (NSG) mice were purchased from the University of California, Los Angeles (UCLA) Medical Center animal-breeding facility and Jackson Laboratories at 6–8 weeks of age. All mice were kept under defined pathogen-free conditions at an animal facility approved by the AAALAC of the Division of Laboratory Animals (DLAM) at UCLA. Mice are housed in a controlled environment with a 12:12 hours light-dark cycle (e.g., lights on at 6:00 AM and off at 6:00 PM). The temperature in the vivarium is maintained between 20–26 °C (68–79 °F). The relative humidity in the environment is regulated between 30–70%. Mice are housed in irradiation-sterilized, ventilated, disposable cages with corn cob bedding, and provided irradiated pico vac diet as feed. Mice were drug and test naïve prior to injection of tumour cells.

### Patient-derived orthotopic xenografts
Intracranial mouse xenograft studies were as previously described[52]. Briefly, GS025 (shControl, shBCL-$X_L$), GS005 (shControl, shBCL-$X_L$), GS025, GBM39, GS027, GS054, GS147, and GS005 cells were transduced with secreted Gaussia luciferase (sGluc)-encoding reporter gene (Prolume Ltd., pLetni_CMV_GLeu_T2A_eGFP plasmid) to enable non-invasive and routine quantification of tumour burden[50] as well as GFP-guided microdissection of the tumour tissue post euthanasia. $2 \times 10^5$ cells were injected into the right striatum of the brain in female NSG mice (6–10 weeks old), 2 mm lateral and 1 mm posterior to the bregma, at a depth of 2 mm. Tumour burden was monitored based on 1-2x weekly measurements of secreted Gaussia luciferase, once tumours entered an exponential growth rate, mice were randomised into treatment arms, there were no covariates for which a control was needed. Sample sizes were chosen based on estimates from pilot experiments. Endpoints were determined primarily by body conditioning score, especially focusing on the 30% weight loss threshold, decreased mobility, uncontrollable seizures and/or bleeding, and respiratory distress. Other criteria under the ARC policy on Humane Treatment and Endpoints was also assessed – tumour burden not evaluated as euthanasia criteria. Investigators were not blinded to group allocation or assessment of outcome. For WES and RNA-seq analysis as well as ex vivo BH3 profiling, tumour cells were purified via mechanical and enzymatic dissociation using the Miltenyi Biotec Mouse tumour dissociation kit (130-096-730). Antibody-conjugated magnetic beads removed myelinated and mouse cells (Miltenyi, 130-104-694) in column-based filtration steps.

## Whole-exome and RNA sequencing

The exome capture was performed by using either SeqCap EZ Human Exome Library V3 kit or KAPA HyperExome enrichment kit according to the manufacturer's instructions. Transcriptome libraries were constructed from poly(A) selected RNA using the Nugen Universal Plus mRNA-seq library prep kit. Paired-end short reads (150 bp) were sequenced on the Illumina HiSeq 3000 or Novaseq 6000 platforms.

## Whole-exome QC and mapping

To prepare read alignments for analysis, all sequencing data were passed through Cutadapt[61] (v2.8). To remove any reads due to mouse cell contamination, we used the BBsplit function of the Bbtools package[62] to map short reads to human (GRCh38) and mouse (mm10) genomes simultaneously; only reads that can map to human genome uniquely were kept. Mapping (BWA 0.7.17-r1188[63]), marking duplicates, and recalibrating the base quality scores were performed based on the human genome (GRCh38) according to the pre-processing workflow (GATK v4.2.0.0[64]) of GATK Best Practices. Mean coverage for exome data was 100 - 150x for both tumour and normal.

## RNA-seq QC and mapping

The Seal program from Bbtools was used to identify and remove reads unambiguously aligning to the mouse transcriptome (vM22). Filtered reads were then processed through the UCSC Toil RNA sequencing pipeline[65] as described previously[66]. Transcripts per million (TPM) expression value outputs from alignment with STAR (v2.4.2a)[67] and quantification with RSEM (v1.2.25)[68] were used for downstream analysis.

## Somatic mutation and copy number alteration calling

For tumour samples with a sequenced matched normal, Mutect2 (v4.2.0.0)[69], MuSE v1.0rc[70], and Varscan2[71] were used to call SNPs, small insertions and deletions. Only variants with a minimum coverage of 20 reads and identified by at least two mutation callers were selected for further analysis. For samples lacking a matched normal, variant calling was performed using Mutect2 in tumour-only mode. Variants were subsequently compared to the matched normal sample when available and a constructed panel of normal samples following GATK best practices. The identified variants underwent filtering based on their occurrence frequency in the COSMIC database[72] (with a threshold of more than seven occurrences in CNS tumours) or their annotation as "likely oncogenic" or "oncogenic" according to OncoKB[73]. The CNVkit package was used to detect copy number changes from whole-exome sequencing data[74].

Identification and allelic frequency estimation of Gervin variants was derived from the alternative transcript splicing of EGFR detected in RNA sequencing data. The determination of EGFRvIII calls involved calculating the fraction of reads mapping to two specific junctions: one between exons 7 and 8 and the other corresponding to the aberrant junction between exons 1 and 8. Sample with EGFRvIII transcript allele frequencies (TAFs) over 10% were considered positive variant.

## Clinical characteristics

Information reported in Supplementary Table 1, along with 1p/19q codeletion and MGMT methylation is from Foundation of Medicine and clinical pathology reports. Clinical reports containing MGMT methylation status was unavailable for HK336 and GBM39. For these samples, MGMT status was determined through bisulfite sequencing as previously described[44]. GBM39 clinical characteristics were previously published[75].

## MGMT status

MGMT status was determined using both MGMT methylation[76] and MGMT RNA expression (median of MGMT RNA expression from all samples used to define cut-off between high and low expression)[77]. Tumours with either methylation or low expression were deemed MGMT negative; tumours unmethylated with high MGMT expression were designated MGMT positive.

## GTEx and TCGA samples public datasets

Gene expression data from glioblastoma tumours and normal frontal cortex brain samples were obtained from the Toil RNAseq Recompute[65] project for TCGA GBM[31] and GTEx[78] respectively, publicly available through UCSC XenaBrowser[79].

## Reagents and antibodies

Chemical inhibitors from the following sources were dissolved in DMSO for in vitro studies: A-1155463 (ChemieTek, CT-A115), S63845 (ChemieTek, CT-S63845), ABT-199 (ChemieTek, CT-A199), Temozolomide (Selleck Chemicals, S1237), Navitoclax (MedChemExpress, HY-10087), A-1331852 (Abbvie), Z-VAD-FMK (Selleck Chemicals, S7023), Pifithrin-α (PFTα) HBr (Selleck Chemicals, S2929), O6-Benzylguanine (Selleck Chemicals, S3658), nutlin-3A (Selleck Chemicals, S8059). ABBV-155 (Abbvie-MTA) suspended in sterile saline. The following antibodies were obtained from the indicated sources and used for immunoblotting: anti-β-actin (8H10D10) mouse mAb (Cell Signaling, 3700), anti-α-tubulin (DM1A) mouse mAb (Cell Signaling, 3873), anti-p53 (DO-1) mouse mAb (Santa Cruz Biotechnology, SC-126), anti-BAX (D2E11) rabbit mAb (Cell Signaling, 5023), anti-BAK rabbit mAb (Cell signaling, 3814), anti-BIM (C34C5) rabbit mAb (Cell Signaling, 2933), anti-BID (Human specific) rabbit mAb (Cell signaling, 2002), anti-PUMA (D30C10) rabbit mAb (Cell Signaling, 12450), anti-Noxa (D8L7U) rabbit mAb (Cell Signaling, 14766), anti-Bcl-2 (50E3) rabbit mAb (Cell Signaling, 2870), anti-BCL-X$_L$ (54H6) rabbit mAb (Cell Signaling, 2764), anti-MCL-1 (D35A5) rabbit mAb (Cell Signaling, 5453), anti-HRK (PRS3771) rabbit mAb, and anti-cytochrome $c$ rabbit mAb (Cell Signaling, 4272). Antibodies used for immunoprecipitation were obtained from the following sources: anti-BCL-X$_L$ (54H6) rabbit mAb (Cell Signaling, 2764), anti-MCL-1 (D35A5) rabbit mAb (Cell Signaling, 5453). Secondary antibodies were obtained from the following sources: anti-rabbit IgG HRP-linked (Cell Signaling, 7074) and anti-mouse IgG HRP-linked (Cell Signaling, 7076). For B7-H3 immunohistochemistry was obtained from R&D Systems (AF1027) and used at concentration of 1:1000. All immunoblotting antibodies were used at the antibody manufacturers' specifications. Immunoprecipitation antibodies were diluted according to the manufacturer's instructions (1:200 for MCL-1 and BCL-X$_L$). Secondary antibodies were used at a dilution of 1:5000.

## Immunoblotting

Cells were lysed in RIPA buffer (Boston BioProducts) with Halt™ Protease and Phosphatase Inhibitor (Thermo Fisher) and were subsequently centrifuged at 14,000 g for 15 min at 4 °C. Protein samples were then heated at 80 °C with NuPAGE LDS Sample Buffer (Thermo Fisher) and NuPAGE Sample Reducing Agent (Thermo Fisher) and separated using SDSPAGE on 12% Bis-Tris gels (Thermo Fisher) and transferred to nitrocellulose membrane (GE Healthcare). Immunoblotting was performed as stated. Membranes were developed using the SuperSignal™ system (Thermo Fisher) and imaged using the Odyssey Fc Imaging System (LI-COR). Signal quantification was performed using the Image Studio™ software (LI-COR). Expression of each BCL-2 family protein determined in relation to the loading control (Actin or Tubulin). Fold change calculated relative to DMSO treated control.

## Synergy score calculations

Performed by incubating 1500 cells per well in 384-well plates for 48 h with BH3 mimetics (A-1155463, S63854, ABT-199). A 7-point titration curve of each drug was performed in triplicate. Cell Titer Glo Luminescent Cell Viability Assay (Promega) was used to measure cell viability from control of each drug. Luminescence (integration time 1 s)

was recorded on a CLARIOstar microplate reader (BMG Labtech). Zip synergy scores calculated using Synergy Finder[80].

### Annexin V apoptosis assay

Cells were collected and analyzed for annexin V and PI staining according to the manufacturer's protocol (BD Biosciences). Briefly, cells were plated at $5 \times 10^4$ cells/mL and treated with the appropriate drugs. At the indicated time points, cells were collected, trypsinized, washed with PBS, and stained with annexin V and PI for 15 min. Samples were then analyzed with a BD LSRII or Attune flow cytometer.

### Senescence

Senescence was measured using the CellEvent™ Senescence Green Detection Kit (Thermo Fisher, C10850). Cells were transduced with secreted Gaussia luciferase (sGluc)-encoding reporter gene (Prolume Ltd., pLetni_CMV_GLeu_T2A_eGFP plasmid), edited to express pDECKO-mCherry commercial backbone from Addgene instead of eGFP, to enable florescent imaging of cells. The cells were quantified via the Incucyte Live-Cell Imaging Analysis System (Sartorius). Representative images were taken with the EVOS M5000 (Thermo Fisher).

### Immunoprecipitation

Cells were collected, washed once with PBS, and incubated in IP lysis buffer (2% CHAPS) at 4 °C for 15 min. 300–500 µg of each sample was then precleared in Protein A/G Plus Agarose Beads (Thermo Fisher) for 1 hr. After preclearing, samples were then incubated with antibody–bead conjugates overnight according to the manufacturer's specifications. The samples were then centrifuged at 1000 g for 1 min, and the beads were washed with 500 µL of IP lysis buffer five times. Proteins were eluted from the beads by boiling in 2× LDS Sample Buffer (Invitrogen) at 95 °C for 5 min. Samples were analyzed by immunoblotting as described above.

### BH3 profiling

Cells were disassociated into single-cell suspensions and resuspended in MEB buffer (150 mM mannitol, 10 mM HEPES-KOH, 50 mM KCl, 0.02 mM EGTA, 0.02 mM EDTA, 0.1% BSA, and 5 mM succinate). 50 µL of cell suspension ($3 \times 10^4$ cells/well) was plated in wells holding 50 µL MEB buffer containing 0.002% digitonin and the indicated peptides in 96-well plates. Plates were then incubated at 25 °C for 50 min. Cells were then fixed with 4% paraformaldehyde for 10 min and neutralized with N2 buffer (1.7 M Tris and 1.25 M glycine, pH 9.1) for 5 min. Samples were stained overnight with 20 µL of staining solution (10% BSA and 2% Tween 20 in PBS) containing DAPI and anti-cytochrome $c$ (Bio Legend clone 6H2.B4, cat. No. 612310). The following day, cytochrome $c$ release was quantified with a BD LSRII or Attune flow cytometer. Measurements were normalized to negative control (DMSO). All peptide conditions run with two biological replicates.

### Ex vivo BH3 profiling

Tumour cells were purified as described above and resuspended in MEB buffer and BH3 profiling immediately preformed.

### Dynamic BH3 profiling

Cells were treated for 48 hours with either vehicle, TMZ (50 µM), IR (5 Gy) or ABBV-155 (1 µg/mL – 72 hours), and then BH3 profiling was performed as described above. For each peptide the change in cytochrome $c$ release was calculated by subtracting the vehicle from treated conditions to determine how apoptotic dependencies shift with treatment.

### Dynamic ex vivo BH3 profiling

Mice bearing PDX025 and PDX005 tumour were anesthetized with ketamine/xylazine at 1.25 mg/kg and cranially irradiated with 10 Gy IR.

After 48 hours mice were euthanized, and tumour cells were purified and used for BH3 profiling as described above. PDX039 and PDX147 were treated with ABBV-155 (10 mg/kg, $qw$, i.p.). After 10 days mice were euthanized, and tumour cells were purified and used for BH3 profiling as described above. For each peptide the change in cytochrome $c$ release was calculated by subtracting the vehicle tumour cells from treated tumours to determine how apoptotic dependencies shifted with treatment.

### Immunohistochemistry

Immunohistochemistry was performed on 4µm sections cut from FFPE (formalin-fixed, paraffin-embedded) blocks. Sections were then deparaffinized with xylene and rehydrated through graded ethanol. Antigen retrieval was achieved with a pH 9.5 Nuclear Decloaker (Biocare Medical) in a decloaking pressure cooker at 95 °C for 40 min. Tissue sections were then treated with 3% hydrogen peroxide (lot 161509; Fisher Chemical) and with Background Sniper (Biocare Medical) to decrease nonspecific background staining. Primary anti-B7-H3 was applied in a 1:100 dilution for 80 min, and detection was then performed with a MACH 3 Rabbit HRP-Polymer Detection kit (Biocare Medical). Visualization was achieved with VECTOR NovaRED (SK-4800; Vector Laboratories) as a chromogen. Finally, sections were counterstained with Tacha's Automated Hematoxylin (Biocare Medical).

### Genetic manipulation

HEK-293-FT cells (ATCC) were transfected using lipofectamine 2000 (Invitrogen) to produce lentiviruses for genetic manipulation, which were collected 48 hours after transfection. Short hairpin RNAs (shRNAs) against PUMA and BCL-$X_L$ were purchased from Sigma (shPUMA1: TRCN0000033610, shPUMA2: TRCN0000033612. shBCL-$X_L$1: TRCN0000033499, shBCL-$X_L$2: TRCN0000033500, shCtl: SHC002). TP53 CRISPR-Cas9 gene disruption was performed with the LentiCRISPR V2 vector (Addgene, 52961) using the following oligonucleotide sequences: sgp53-1 CCGGTTCATGCCGCCCATGC, sgp53-2 GAGCGCTGCTCAGATAGCGA, sgControl GTAATCCTAGCACTTT-TAGG. Cells were spinfected with lentivirus and Polybrene (1 ug/mL) at 800 g for 1.5 hours (32 °C). Cells were immediately transferred into standard media and subjected to puromycin selection (1 mM puromycin) after 5 days of recovery.

### Secreted Gaussia luciferase measurements

To measure tumour burden via the levels of sGluc, blood (6 µL) was collected from the tail vein and mixed with 50 mM EDTA immediately to prevent coagulation. Chemiluminescence was measured after injection of 100 µL of 100 µM coelentarazine (Nanolight) in a 96-well plate to obtain sGluc activity, as described before[50].

### Mouse treatment studies

GS025 (shControl, shBCL-$X_L$), GS005 (shControl, shBCL-$X_L$) cells were injected as described above. Three days post injection mice were anesthetized with ketamine/xylazine at 1.25 mg/kg and cranially irradiated with 10 Gy IR. Mice bearing PDX005, PDX025, PDX027, PDX054, PDX039 and PDX147 were monitored for exponential tumour growth by sGluc measurements. Once aggressive growth was achieved, mice were randomised and treated with vehicle, 10 mg/kg ABBV-155 IP qw for three weeks, 10 Gy IR qw for two weeks, or ABBV-155 and IR. ABBV-155 was suspended in sterile saline. For IR treatments, mice were anesthetized with ketamine/xylazine at 1.25 mg/kg. Mice were humanely euthanized with CO2 when endpoints were reached.

### Comparing the predictive abilities of BIM$^{AUC}$ and multi-omics feature sets

A method that performs Pearson correlation followed by the elastic net regression[81] was used to evaluate the functionally defined feature,

BIM^{AUC} AUC, and multi-omics feature sets (transcriptomics (RNA), copy number alterations (CNA), and mutations (MUT)) to predict the response to the dual treatment of TMZ + BCL-X$_L$i or IR + BCL-X$_L$i when *TP53* mutation, MGMT status (a binary metric based on both expression and methylation) and *MDM2* amplification (Global Molecular (GM) Model) are known. As the elastic net may cause the estimates of non-zero coefficients to be biased towards zero[82], a relaxed elastic net, which takes the selected variables by the elastic net and refits an unpenalized linear model, was used in place of the originally described elastic net in the abovementioned model building algorithm. Elastic net was performed using glmnet R package (v4.1-3)[83]. Both BIM^{AUC} and multi-omics features interact with *TP53, MDM2* and MGMT through various ways to create the starting feature sets (Supplementary Table 2).

To reduce the collinearity in the elastic net, we merge genes with exactly the same values across all samples (RNA expression $p = 17{,}761$, Copy number alterations $p = 1257$, mutations $p = 10$) followed by interactions and additions between these features in certain models.

Starting feature sets are displayed in Supplementary Table 2. Top features in the table are based on the absolute Pearson correlation between each feature and the response variable. Number of combinations is fewer than the full combinations due to certain products between features resulting in zero variance across samples and are excluded in the starting feature set.

Cell death data were acquired after TMZ or IR with BCL-X$_L$i treatment for both the training cohort ($n = 26$) and an independent verification cohort ($n = 12$). The training cohort was divided into training, validation, and test subsets (Fig. 4A) in order to perform nested cross validation (CV). The inner loop of nested CV performs the hyperparameter tuning for Pearson correlation cutoff, elastic net (ridge[84] and lasso[85] percentage alpha, penalizing strength lambda and debiasing factor gamma using train and validation subsets (5-fold CV with 20 repeats). The outer loop is a leave-one-out CV (LOOCV) (test data has $n = 1$ for 26 folds), which was designed to estimate the model building performance by fitting the best parameter sets to the entire training plus validation data and then predicting the test data. The predictive accuracy of the models was evaluated by calculating the Root Mean Square Error (RMSE) of the sample not used for training during LOOCV as the outer-loop CV[85]. In addition, the whole training cohort is also fitted using the best parameter sets derived from the same inner tunning algorithm to predict the testing cohort as a second validation for the modeling performance. Both nested CV and testing errors were quantified by RMSE. Model parameters were fitted on the entire training cohort prior to predicting the independent verification cohort.

In Supplementary Data Figs. 5A, 6C, ∗ represents simple products between variables without individual main effects.

### Identifying the most important features in the combined feature set

As a second method to test if features in the IMF interaction are more correlative than multi-omics GM features when fitting to TMZ + BCL-X$_L$i or IR + BCL-X$_L$i response, lasso was performed on the combined feature set IMF + GM. By increasing the penalty strength in LASSO, we assigned an order to the features based on how late their coefficient is shrunken to zero. The later the shrinkage to zero, the more valuable the feature is at predicting cell death. To assess if the number of IMF features within the top 100 features is beyond expectation, the relationships between TMZ + BCL-X$_L$i or IR + BCL-X$_L$i response and feature sets were shuffled 1000 times to calculate permutation p-values.

### GBM Apoptotic Vulnerability Assessment (GAVA)

Samples were scored as GAVA positive or negative by applying the top ranked IMF features (IR + BCL-X$_L$i = BIM^{AUC}*TP53*MDM2; TMZ + BCL-

X$_L$i = BIM^{AUC}*TP53*MDM2*MGMT) as binary variables to score the gliomaspheres. Cutoff for high and low priming as determined using the ROC curves in Supplementary Fig. 4D. Cut off for response or non-response to IR or TMZ with BCL-X$_L$i is taken from the mean of each group (Fig. 4A).

### Quantification and statistical analysis

Unless otherwise specified, comparisons were made with two-tailed unpaired Student's t-tests, and P values < 0.05 were considered statistically significant. All data from multiple independent experiments were assumed to be of normal variance. For each experiment, replicates are noted in the figure legends. Data represents mean ± s.d. values unless otherwise indicated. All statistical analyses were calculated in Prism 10.0 (GraphPad). The code for elastic net and cross validations was written in R (v4.1.2). For all in vitro and in vivo experiments, no statistical method was used to predetermine sample size, and no samples were excluded. For in vivo tumour measurements, students t-test were used to compare between groups. As described above, all mice were randomised before studies. For all figures: $p > 0.05$ = ns; $p < 0.05$ = *; $p < 0.01$ = **; $p < 0.001$ = ***; $p < 0.0001$ = ****.

### Reporting summary

Further information on research design is available in the Nature Portfolio Reporting Summary linked to this article.

## Data availability

The whole-exome and RNA sequencing data generated in this study have been deposited in the dbGap database under accession code phs003286. Sequencing for samples used in this paper are marked with variable "FERNANDEZ_2024". The remaining data are available within the Article, Supplementary Information or Source Data file.

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

## Acknowledgements

The authors thank Chris Tse and all the Nathanson Lab SRAs who have contributed to building our glioma sample and model library. We thank Vida Zhang for helping with the synergy experiments and Raymond Lim for helping with assessment of BCL-2 family expression. We thank Dr. Harvey Herschman for his advice and thoughts on the manuscript. We thank all past and present members of the Nathanson Lab for their valuable critiques and discussions. We wish to acknowledge Dr. Xinmin Li at the Center for Genomics and Bioinformatics (TCGB) for sequencing supervision. All graphics and illustrations created with Biorender (Biorender.com). Support was provided by the Sheila and Stanford L. Kurland Family Foundation (T.F.C., D.A.N.), National Brain Tumor Society (D.A.N., E.G.F., T.F.C.), Uncle Kory Foundation (D.A.N., T.F.C.), Ziering Family Foundation (D.A.N., T.F.C.), The Bloomfield Foundation (D.A.N., T.F.C.), Harry Zimmerman in memory of his wife Sandra Zimmerman (D.A.N., T.F.C.), Art of the Brain in memory of Judi Kaufman (D.A.N., T.F.C.), a generous gift from Deborah Newman Sharpe (D.A.N., T.F.C.) and the National Institutes of Health grants P50CA211015 (D.A.N., T.F.C., L.M.L., T.G.G.) and R01CA227089 (D.A.N., T.G.G., T.F.C., L.M.L.). E.G.F. is a predoctoral fellow supported by the UCLA Tumor Cell Biology Training Program (US HHS Ruth L. Kirschstein Institutional National Research Service Award# T32 CA009056). Finally, we are grateful to the patients with brain cancer and their families who consented to donating tumour tissue for this research.

## Author contributions

Conceptualization: E.G.F., W.X.M., and D.A.N.; Resources: E.G.F., L.M.L., A.J.S., T.F.C. and D.A.N.; Data curation: N.A.B., H.Z., and C.L.A.; Software: K.S., N.A.B., and H.Z.; Formal analysis: E.G.F., W.X.M., K.S., N.A.B., J.K. and C.L.A.; Supervision: G.L., W.H.Y., J.J.L., T.G.G., T.F.C and D.A.N.. Funding acquisition: E.G.F., L.M.L, T.G.G., T.F.C., and D.A.N.; Validation: EGF, WXM, and KS. Investigation: E.G.F., W.XM., K.S., M.P., and P.Y.; Visualization: E.G.F., W.X.M., K.S., and D.A.N.; Methodology: E.G.F., W.X.M., K.S., N.A.B., J.K., H.Z., G.L., W.H.Y., A.J.S., J.J.L., T.G.G., T.F.C., and D.A.N.; Writing-original draft: E.G.F., W.X.M., and D.A.N.; Project

administration: E.G.F., W.X.M., and D.A.N.; Writing-review and editing: EGF, K.S., N.A.B., D.C., T.G.G., D.A.N.

## Competing interests

The authors declare the following competing interests, D.A.N. is a co-founder of Trethera Corporation and has equity in the company. D.A.N. and T.F.C. are co-founders of Katmai Pharmaceuticals and have equity in the company. T.G.G. has consulting and equity agreements with Auron Therapeutics, Boundless Bio, Coherus BioSciences and Trethera Corporation. E.G.F., W.X.M., K.S., N.A.B., J.K., H.Z., M.P., P.Y., C.L.A., D.C., L.M.L., G.L., W.H.Y., F.J.R., S.J.D., A.J.S., and J.J.L. have no competing interests to declare.

## Additional information

[1]Department of Molecular and Medical Pharmacology, David Geffen School of Medicine, University of California Los Angeles, Los Angeles, CA 90095, USA. [2]Department of Bioengineering, University of California, Los Angeles, Los Angeles, CA, USA. [3]Department of Biostatistics, Jonathan and Karin Fielding School of Public Health, Los Angeles, California, USA. [4]Jonsson Comprehensive Cancer Center, University of California, Los Angeles, Los Angeles, CA, USA. [5]Department of Neurosurgery, David Geffen School of Medicine, University of California, Los Angeles, Los Angeles, CA, USA. [6]Department of Pathology, David Geffen School of Medicine, University of California, Los Angeles, Los Angeles, CA, USA. [7]Department of Biology, Stanford University, Stanford, CA 94305, USA. [8]AbbVie, Inc., 1 North Waukegan Road, North Chicago, IL 60064, USA. [9]Department of Human Genetics, University of California, Los Angeles, CA 90095-7088, USA. [10]Department of Computational Medicine, University of California, Los Angeles, CA 90095-1766, USA. [11]Department of Statistics and Data Science, University of California, Los Angeles, CA 90095-1554, USA. [12]UCLA Metabolomics Center, University of California Los Angeles, Los Angeles, CA 90095, USA. [13]Crump Institute for Molecular Imaging, University of California Los Angeles, Los Angeles, CA 90095, USA. [14]Department of Neurology, David Geffen School of Medicine, University of California, Los Angeles, Los Angeles, CA, USA. ✉e-mail: dnathanson@mednet.ucla.edu

