## [Peer Review file · Nature Communications]

Integrated molecular and functional characterization of the intrinsic apoptotic machinery identifies therapeutic vulnerabilities in malignant glioma

Corresponding Author: Dr David Nathanson

Version 0:

Reviewer comments:

Reviewer #1

(Remarks to the Author)

The manuscript by Fernandez and colleagues elegantly combines molecular and functional analyses to identify new treatments to better treat malignant glioma. The study is well presented and comprehensive, and explores how to overcome anti-apoptotic adaptations to standard therapy using ABBV-155. These results could potentially impact the clinic in the future. However, a few elements of this manuscript could be improved.

- Figure 1G is not mentioned in the text
- The expression of HRK, a sensitizer that specifically binds to BCL-xL, is not analyzed.
- When cells are exposed to IR senescence can be induced. Have the authors checked for senescence markers?
- When performing ROC curve analyses, the AUC value should be clearly indicated.
- Overall, when performing dynamic BH3 profiling analyses, the authors consider the area under the BIM curve. It would be interesting to also evaluate $\Delta\%$ priming as originally described (meaning the highest difference in priming when comparing two curves).

I would like to congratulate the authors for such an excellent study.

Reviewer #2

(Remarks to the Author)

This is a very interesting study evaluating the potential for functional assessment of intrinsic apoptotic priming on response to therapy in GBM. The authors convincingly demonstrate that a significant subset of GBM are primed to undergo apoptosis via MCL1 and/or BCL-XL dependency. There are clear translational ramifications of this finding, and they nicely demonstrate this in the final figures using an ADC with a BCLXL inhibitory warhead. Overall, the studies are very carefully performed, results are clearly described, and the conclusions are nicely supported by the data provided. This is an important manuscript that clearly moves the field forward by highlighting the potential of functional assessment of apoptotic pathways in a precision medicine paradigm. This manuscript will be of high interest to those in the Neuro-oncology field, but also in oncology therapeutics and precision medicine in general. There are only very minor editorial issues that should be addressed.

Minor issues:

1. In Figure 2H - the x-axis of the not labeled and the reader has to infer from the legend that these are measurements over time. Is there a way to more clearly label the figure itself?
2. In Figure 4B/D - within the legend or figure, the authors should specify the significance of the darker shading as it relates to figure 4A. This may not be readily apparent to all readers.
3. The authors should rephrase the description of the in vivo results to avoid using the term 'synergy/synergistic'. In survival/efficacy studies, this term is typically reserved for describing a specific statistical analysis for synergy.

Reviewer #3

(Remarks to the Author)

Fernandez et al have conducted a thorough study of malignant glioma. Analysis of freshly isolated glioma patient sample using integrated molecular and functional BH3 profiling and utilizing machine learning was established, and a predictive biomarker (GAVA) helps to target glioma intrinsic apoptosis. The study shows that anti-apoptotic proteins MCL-1 and BCL-XL protect glioma tumor cells against cell death. The results were obtained using glioma patient samples and derivative models. The presented approach allows prediction of therapy-induced cell death and suggests a new treatment approach for glioma patients.

The study is well planned and conducted. It has the potential to advance the treatment of glioma patients. Similar methodology could be applied to other tumor types as well. The methods used have largely been published earlier.

The work supports the conclusions and claims.

Data analysis, interpretation and conclusions are appropriate.

The methodology is sound and meets the standards in the field.

The methods are described in detail, and work could be reproduced.

Specific comments:

Limitations of the study should be discussed by the authors. BH3 profiling requires only short ex vivo culture of the cancer cells and thus minimizes adaptive changes which occur during prolonged culture. Yet, the cancer cells are cultured ex vivo after removal of many other cell types which may trigger changes in the cells. This should be discussed in more detail.

Language check should be performed. Abbreviations need to be spelled out (e.g. line 183 MOMP). Line 612: is some text missing? References for some methods are missing (e.g. line 746). Figure 1 panel B: Labels at the top are slightly misaligned. Figure 1 legend A: modify "characterization of both glioma patient samples (n=31)". Extended figure 2: panel C appears to miss some data according to the legend.

We thank all the reviewers for their insightful and constructive comments, and the overall enthusiasm for the findings and translational potential of the work. We have endeavoured to address the comments and critiques raised by each of the reviewers.

****REVIEWER REPORT****

Reviewer #1 – Apoptosis, precision oncology

The manuscript by Fernandez and colleagues elegantly combines molecular and functional analyses to identify new treatments to better treat malignant glioma. The study is well presented and comprehensive and explores how to overcome anti-apoptotic adaptations to standard therapy using ABBV-155. These results could potentially impact the clinic in the future. However, a few elements of this manuscript could be improved.

We wish to thank the reviewer for his/her constructive feedback on the manuscript and appreciate that the reviewer found the work to be comprehensive and having potential for clinical impact.

- Figure 1G is not mentioned in the text

Thank you for identifying this error. We have fixed the text so that Figure 1G is mentioned.

- The expression of HRK, a sensitizer that specifically binds to BCL-xL, is not analysed.

In the initial version of the manuscript, we analysed HRK expression by RNA sequencing which indicated HRK is lowly expressed at basal conditions in GBM tumours (Supplemental Fig. 1C).

In new data shown below and in the revised manuscript, we show that with 48 hr of TMZ (50µM) or IR (5 Gy) treatment, HRK levels do not consistently change in either p53 WT or mut-p53 gliomaspheres (Supplemental Fig. 3B - top). Moreover, any changes in HRK levels with TMZ or IR treatment were independent of p53 as determined by CRISPR-Cas9 mediated knockout of TP53 (Supplemental Fig. 3C - bottom). These data suggest HRK does not have prominent role in our observed p53-mediated apoptotic block dynamics following IR or TMZ treatment. Note, pancreas tissue was used as a control for HRK expression as it is known to express HRK (Inohara et. al., *The Embro Journal*, 1997).

When cells are exposed to IR senescence can be induced. Have the authors checked for senescence markers?

This is an excellent point raised by the reviewer as treatment-induced senescence has been linked to BCL-X_L dependences in GBM (Rahman et. al., *Mol Cancer Research*, 2022). To assess whether markers of senescence are observed at the time point in which we observed a IR or TMZ-induced switch to a sole dependency on BCL-X_L (48 hr), we treated 2 p53 WT gliomaspheres (GS025 and GS187) with 5 Gy IR for 48 hr (dose and time used in manuscript) and then assessed senescence with the CellEvent™ Senescence Green Detection Kit - a fluorescent based reagent that contains two galactoside moieties, making it specific to β-galactosidase. We observed no detectable increase in senescence with treatment relative to vehicle. By contrast, treating gliomaspheres with a dose (15 Gy) and a time (6 days) previously linked to IR-induced senescence could significantly increase markers of senescence in our GBM spheres. These data lead us to conclude that the induction of senescence is not necessary for the treatment-induced dependency on BCL-X_L in patient derived GBM cells. This figure is included as Supplementary Fig. 2D and E.

- When performing should be clearly indicated.

ROC curve analyses, the AUC value

To address this point, we added the AUC value to the ROC graphs in Supplemental Figure 4D.

- Overall, when performing dynamic BH3 profiling analyses, the authors consider the area under the BIM curve. It would be interesting to also evaluate $\Delta\%$ priming as originally described (meaning the highest difference in priming when comparing two curves).

We did not use dynamic BH3 profiling (DBP) for assessment of GBM cell priming. Rather, the DBP assay was employed exclusively to assess changes in apoptotic block dependencies by using HRK, MS1, or ABT-199 peptides (Fig 2). Apoptotic priming (via AUC of BIM peptide titrations) was used only to assess the basal primed state of the GBM cells. We apologize if this was not clear and can revise the main text if necessary.

I would like to congratulate the authors for such an excellent study.

Much appreciated

Reviewer #2 – Glioblastoma

This is a very interesting study evaluating the potential for functional assessment of intrinsic apoptotic priming on response to therapy in GBM. The authors convincingly demonstrate that a significant subset of GBM is primed to undergo apoptosis via MCL1 and/or BCL-XL dependency. There are clear translational ramifications of this finding, and they nicely demonstrate this in the final figures using an ADC with a BCLXL inhibitory warhead. Overall, the studies are very carefully performed, results are clearly described, and the conclusions are nicely supported by the data provided. This is an important manuscript that clearly moves the field forward by highlighting the potential of functional assessment of apoptotic pathways in a precision medicine paradigm. This manuscript will be of high interest to those in the Neuro-oncology field, but also in oncology therapeutics and precision medicine in general. There are only very minor editorial issues that should be addressed.

We thank the reviewer for his/her comments regarding the execution of the studies and the importance of this work in oncology and precision medicine. We also appreciate the reviewer for pointing out the minor issues that were missed at first submission.

Minor issues:

1. In Figure 2H - the x-axis of the not labelled and the reader must infer from the legend that these are measurements over time. Is there a way to more clearly label the figure itself?

We apologize this was not clear in the originally figure. To address this, we added a “Days of treatment” label for the x axis as well as arrows indicating the time progression for the individual groups. We hope this makes the figure easier to interpret.

2. In Figure 4B/D - within the legend or figure, the authors should specify the significance of the darker shading as it relates to figure 4A. This may not be readily apparent to all readers.

Thank you for the suggestion. We added this sentence “Darker coloring signifies responders identified in 4A” to the legends of Figure 4B/D.

3. The authors should rephrase the description of the in vivo results to avoid using the term 'synergy/synergistic'. In survival/efficacy studies, this term is typically reserved for describing a specific statistical analysis for synergy.

We completely agree with the reviewer. We have removed the term synergy/synergistic when unable to confirm statistically.

Reviewer #3 – Computations, integrative omics analysis

Fernandez et al have conducted a thorough study of malignant glioma. Analysis of freshly isolated glioma patient sample using integrated molecular and functional BH3 profiling and utilizing machine learning was established, and a predictive biomarker (GAVA) helps to target glioma intrinsic apoptosis. The study shows that anti-apoptotic proteins MCL-1 and BCL-XL protect glioma tumor cells against cell death. The results were obtained using glioma patient samples and derivative models. The presented approach allows prediction of therapy-induced cell death and suggests a new treatment approach for glioma patients.

The study is well planned and conducted. It has the potential to advance the treatment of glioma patients. Similar methodology could be applied to other tumor types as well. The methods used have largely been published earlier.

The work supports the conclusions and claims.

Data analysis, interpretation and conclusions are appropriate.

The methodology is sound and meets the standards in the field.

The methods are described in detail, and work could be reproduced.

We appreciate the reviewer's comments regarding the rigor and execution of the study and for identifying areas within the text that could be improved.

Specific comments:

Limitations of the study should be discussed by the authors. BH3 profiling requires only short ex vivo culture of the cancer cells and thus minimizes adaptive changes which occur during prolonged culture. Yet, the cancer cells are cultured ex vivo after removal of many other cell types which may trigger changes in the cells. This should be discussed in more detail.

We agree with the reviewer's suggestion to add in additional limitations/caveats, in particular how the isolation of tumour cells from neighbouring cells in the TME and/or mechanical digestion of tissue could influence apoptotic characteristics of the tumours that were measured (e.g., BH3 profiling, RNA sequencing, etc). We have modified the text to incorporate this caveat. We have also tried to include other limitations of this work including the uncertainty of ABBV-155 penetration in GBM patient tumours and the unknown impact of other drug perturbations on dual apoptotic blocks in GBM.

Language check should be performed. Abbreviations need to be spelled out (e.g. line 183 MOMP). Line 612: is some text missing? References for some methods are missing (e.g. line 746). Figure 1 panel B: Labels at the top are slightly misaligned. Figure 1 legend A: modify "characterization of both glioma patient samples (n=31)". Supplemental figure 2: panel C appears to miss some data according to the legend.

MOMP – spelled out earlier

Thank you for identifying these errors in the text. For Line 612, a period was missing. This is now included. For Line 746, we have added in the appropriate reference. We have also fixed the noted figures and legends and spelled out "MOMP" earlier where it was referenced. Some other minor errors in the text were identified during our revision and were also corrected.